

# Proposed mechanisms of action of herbal drugs and their biologically active constituents in the treatment of coughs: an overview

Jana Pourova[1], Patricia Dias[1], Milan Pour[2], Silvia Bittner Fialová[3], Szilvia Czigle[3], Milan Nagy[3], Jaroslav Tóth[3], Viktória Lilla Balázs[4], Adrienn Horváth[5], Eszter Csikós[4], Ágnes Farkas[4], Györgyi Horváth[4] and Přemysl Mladěnka[1]

[1] Department of Pharmacology and Toxicology, Faculty of Pharmacy, Charles University Prague, Hradec Králové, Czech Republic
[2] Department of Organic and Bioorganic Chemistry, Faculty of Pharmacy, Charles University Prague, Hradec Králové, Czech Republic
[3] Department of Pharmacognosy and Botany, Faculty of Pharmacy, Comenius University Bratislava, Bratislava, Slovak Republic
[4] Department of Pharmacognosy, Faculty of Pharmacy, University of Pécs, Pécs, Hungary
[5] Department of Pharmaceutical Biology, Faculty of Pharmacy, University of Pécs, Pécs, Hungary

Corresponding author
Jana Pourova,
jana.pourova@faf.cuni.cz

## ABSTRACT

Various medicinal plants find their use in cough treatment, based on traditions and long-term experience. Pharmacological principles of their action, however, are much less known. Herbal drugs usually contain a mixture of potentially active compounds, which can manifest diverse effects. Expectorant or antitussive effects, which can be accompanied by others, such as anti-inflammatory or antibacterial, are probably the most important in the treatment of coughs. The aim of this review is to summarize the current state of knowledge of the effects of medicinal plants or their constituents on cough, based on reliable pharmacological studies. First, a comprehensive description of each effect is provided in order to explain the possible mechanism of action in detail. Next, the results related to individual plants and substances are summarized and critically discussed based on pharmacological *in vivo* and *in vitro* investigation.

## INTRODUCTION

Medicinal plants have been used for centuries, including for the treatment of coughs, although their use for these purposes have not always been supported scientifically. Hence, credible phytochemical and pharmacological studies are highly desirable, since their results pave the way towards the design and development of genuinely rational phytotherapy practice that would be well-established, effective and safe. Cough is a very important protective reflex that also occurs in healthy individuals. This reflex maintains the lower airways being clear and protects them, for example, from inhaled particles or

**Table 1 Principal causes of coughs.**

| Cough type | | Possible causes |
|---|---|---|
| Acute cough | Dry | Viral infection<br>Contact with an allergen<br>Inhalation of a foreign particle<br>Inhalation of irritant gases or powders<br>Rarely even partial spontaneous pneumothorax or small pulmonary embolism |
| | Productive | Bacterial infection<br>COPD exacerbation<br>Less typically an attack of bronchial asthma<br>Rarely acute left ventricular failure with pulmonary edema or *arteria pulmonalis* embolism |
| Chronic cough | Dry | Inhalation of pollutants<br>Chronically foreign particles<br>Bronchial asthma<br>Rhinitis, postnasal drip syndrome<br>Gastroesophageal reflux<br>Cough after infection<br>Carcinoma including metastases<br>ACE inhibitor therapy<br>Habitual cough |
| | Productive | Chronic bronchitis<br>Chronic obstructive pulmonary disease (COPD)<br>Bronchiectasis<br>Cystic fibrosis<br>Some infections (tuberculosis, pulmonary mycosis) |

excessive mucus accumulation. In contrast to this useful physiological function, frequent or severe coughing can produce undesirable effects such as a distress of respiratory muscles, interrupted sleep, urinary incontinence, impaired healing of wounds after surgery, and constitutes a social barrier. Cough can be an indicator of a disease, both of pulmonary and non-pulmonary origin. In these cases, cough represents a non-specific and variable symptom. Cough can be underestimated but also overestimated, and it is difficult to recognize when it becomes pathological. However, it can also be an important clue for a diagnosis. Two main criteria are used to classify cough: the presence of sputum and duration. Thus, the productive cough with sputum expectoration *vs* the dry cough, and the acute cough *vs* the chronic cough (longer than 3 weeks (*Irwin, Curley & French, 1990*)) are distinguished. The principal causes of diverse cough types are summarized in Table 1. Sound knowledge of the most common causes of cough and their categorization can contribute to a more extensive use of phytotherapy. Regardless of it, it is always necessary to examine a severe and/or long-term cough, as well as hemoptysis. Last, but not least, multiple causes of cough in one patient are possible.

The purpose of this review is to provide an update on the current state of knowledge on the phytotherapy of cough from the pharmacological standpoint. This review is intended for professionals in the field, such as pharmacognosists, who are experts at botanical aspects and are interested in extending their knowledge by the pharmacological point of

**Table 2 Principal medicinal plants used to support expectoration or used to suppress coughs and corresponding herbal drugs and main constituents.** All data from ref. *Nagy, Mučaji & Grančai (2017)*.

| Plant | Plant parts, herbal drug | Main constituents |
|---|---|---|
| **Herbs used to support expectoration** | | |
| *Pimpinella anisum* L. | fruit, Anisi fructus<br>essential oil, Anisi aetheroleum | *trans*-anethole, anisaldehyde |
| *Eucalyptus globulus* Labill. | leaf, Eucalypti folium<br>essential oil, Eucalypti aetheroleum | 1,8-cineole (eucalyptol), limonene |
| *Foeniculum vulgare* var. *dulce* Mill.<br>or<br>*Foeniculum vulgare* var. *vulgare* Mill. | fruit, Foeniculi amari fructus<br>fruit, Foeniculi dulcis fructus<br>essential oil, Foeniculi amari fructus aetheroleum | *trans*-anethole, fenchone, estragole<br>*trans*-anethole, fenchone, estragole |
| *Hedera helix* L. | leaf, Hederae folium | hederacosides (*e.g.*, hederasaponin C) |
| *Glycyrrhiza glabra* L.<br>or<br>*Glycyrrhiza uralensis* Fisch.<br>or<br>*Glycyrrhiza inflata* Bat. | root, Liquiritiae radix | glycyrrhizin,<br>glycyrrhizic acid,<br>glycyrrhetinic acid,<br>liquiritigenin,<br>isoliquiritigenin,<br>glabridin |
| *Marrubium vulgare* L. | aerial part, Marrubii herba | premarrubiin, marrubiin |
| *Mentha × piperita* L. | essential oil, Menthae piperitae aetheroleum | menthol, menthone, menthyl acetate |
| *Primula veris* L.<br>or<br>*Primula elatior* (L.) Hill | flower, Primulae flos<br>root, Primulae radix | saponins |
| *Thymus vulgaris* L.<br>or<br>*Thymus zygis* L. | aerial part, Thymi herba<br>essential oil, Thymi aetheroleum | thymol, carvacrol |
| **Herbs used to suppress cough** | | |
| *Althaea officinalis* L. | root, Althaeae radix<br>leaf, Althaeae folium | mucilage |
| *Cetraria islandica* (L.) Acharius | thallus, Lichen islandicus | lichenan,<br>isolichenan |
| *Pelargonium sidoides* DC<br>or<br>*Pelargonium reniforme* Curt. | root, Pelargonii radix | tannins, coumarins |
| *Plantago lanceolata* L. | leaf, Plantaginis lanceolatae folium | mucilage, iridoids |

view. Physicians or pharmacists, dealing with cough phytotherapy, are also among the target readership. Our principal goals include the following: (1) to provide an overview of the possible mechanisms of action participating in herbal cough therapy, (2) to supply the examples of herbal drugs or active substances, in which these effects have been reliably demonstrated, and (3) to mention some of the drawbacks of the research in this field. For the sake of clarity, cough will be shortly introduced first. Next, the description of conventional cough therapies will be given. In the main part of the article, established principles of cough phytotherapy will be discussed in a greater detail. The medicinal plants,

**Figure 1 Chemical structures of important plant constituents (alphabetic order).** *Trans*-anethole (1), anisaldehyde (2), apigenin (3), 1,8-cineole (4), estragole (5), fenchone (6), glabridin (7), glycyrrhetinic acid (8), glycyrrhizic acid (9), glycyrrhizin (10), hederasaponin C (11), isoliquiritigenin (12), lichenan (13), limonene (14), liquiritigenin (15), luteolin (16), marrubiin (17), menthol (18), plantamajoside (19), scopoletin (20), thymol (21).

herbal drugs and their components included in this article are listed in Table 2. Chemical structures of important plant constituents are depicted in Fig. 1.

## SURVEY METHODOLOGY

The PubMed and Science Direct databases were used for the search. The aim of this article was to summarize the current knowledge on the effect of medicinal plants, herbal drugs and its biologically active compounds on cough. First, we reviewed experimental articles dealing with the mechanism(s) of action of medicinal plants used in the treatment of cough. Next we focused on these mechanisms using relevant immunological, pathophysiological and pharmacological papers or books, and described them in a very comprehensive way. The experimental methods used to investigate each of these mechanisms were briefly mentioned. The mechanisms were sorted out by importance. The expectorant and antitussive effects were considered as the principal ones, followed by the (less important) anti-inflammatory and anti-infective effects. For the sake of clarity, certain specific aspects of the anti-inflammatory effects were described separately. Poorly studied or speculative mechanisms of action were not disregarded, either. In the next step, selected medicinal plants and herbal drugs (listed in Table 2) were searched for in the same databases *via* different keywords (name of the plant, cough, particular mechanism such as expectorant, antitussive, anti-inflammatory, cytokine, interleukin, interferon, tumor necrosis factor, cyclooxygenase, lipoxygenase, antioxidant, NF-κB, anti-infective, antibacterial, antiviral, TRP channel, adhesion molecule, hyaluronic acid, receptor, ROS, antioxidant…, *in vivo*, *in vitro*) from 1980 to date. Publications reporting the effect of phytotherapy on the common cold (in which cough is only one of the symptoms), were also included. In a few cases, the data from studies, where cough was not used as a primary experimental model, were used to document certain specific effects of medicinal plants and herbal drugs. This is particularly true for the anti-inflammatory effect (*e.g.*, paw edema or various arthritis models) and regulation of the immune response in general.

We also identified several articles reporting sound botanical and analytical data, but the pharmacological experiments were questionable. This was mainly due to the lack of credible toxicity tests, the use of extremely unrealistic doses and concentrations, and last but not least, absurd pharmacological models (*e.g.*, the lack of the target receptors in the chosen animal species). These articles were therefore excluded from further data search. For the same reason, brief comments on the way of performing scientific research in herbal cough therapy were included in the Conclusions section, with emphasis on the weak points.

### Cough and its conventional therapies

The respiratory system is richly innervated. The cholinergic neurons regulate the airway tone, mucus secretion and vasodilation *via* muscarinic receptors, and the sensory neurons detect the harmful stimuli. Cough is initiated following the activation of the airway sensory nerves, and can be preceded by a sensation of airway irritation such as itching. The specific receptors are located between airway epithelial cells, and detect both physical and chemical irritant stimuli, such as inhaled gases, solid or liquid particles, an excessive mucus, or

mediators of inflammation. The chemosensitive C fibers and mechanosensitive Aδ fibers are the main neural fibers responsible for cough. The former are unmyelinated and slower conducting, the latter myelinated and faster conducting. Probably not all participating sensors are already known. Among them, the transient receptor potential (TRP) channels, the ATP-sensitive P2X receptors, GABA$_B$ receptors, sodium channels, neurokinin receptor 1 or α7 subtype of acetylcholine receptor could be of special importance (*Adcock, 2009*; *Bonvini & Belvisi, 2017*; *Zhou et al., 2011*). Whatever the case, activation of the sensors results in depolarization of the plasma membrane. If the stimulus exceeds a threshold value, the action potential is conducted up the vagal afferent nerves and synapses of the *medulla oblongata* where the reflex response is triggered. As a consequence, the medullary respiratory neurons send a signal through the motor nerves to the larynx, respiratory muscles and diaphragm, which results in the cough. First, a deep inspiration and closure of the glottis and vocal cords occurs. Then, the contraction of the expiratory muscles and re-opening of the glottis follows with a short forceful expiration through the mouth. This way, the irritating stimulus can be moved out of the airways (*Chang, 2006*). Notably, cough may also be voluntary, with no reason for a pharmacological intervention.

Therapy is necessary only when it becomes pathological and/or annoying. To choose an appropriate treatment, correct diagnosis of the cough type is necessary. From the pharmacological standpoint, two different approaches are applied. (1) Expectorants, used for the productive cough therapy, facilitate coughing out. On the other hand, (2) antitussives suppress the dry cough. For example, *N*-acetylcysteine makes mucus less viscous and facilitates expectoration, while dextromethorphan and other agonists of opioid receptors suppress the cough center in the CNS. Importantly, expectorants and antitussives act against each other, and, consequently, their co-administration is generally not suitable.

## The main principles of cough phytotherapy

In contrast to conventional medicines, which are generally clearly assigned either to expectorants or antitussives, cough phytotherapy preparations cannot be so clearly classified, since medicinal plants usually contain a mixture of potentially active compounds. The final therapeutic outcome thus reflects the interplay of various effects that a given herbal drug exhibits. Their mutual potentiation or synergism are possible. In addition, it is well known that the contents of the secondary metabolites in a plant species is not constant, but can vary depending on the cultivar, climate, the composition of soil and similar factors, since all of them influence the metabolic pathways and thereby metabolite production. However, similar to the conventional medicines, expectorant or antitussive effects of herbal drugs are the cornerstones for the cough therapy, although this principal effect can be supported by others, for example anti-inflammatory or antibacterial activities. These accompanying effects are often useful both in promoting the expectoration and in relieving the dry cough. Thus, some medicinal plants, herbal drugs and natural substances cannot be categorized in a simple way. For example, *Glycyrrhiza* spp. and its main constituents studied in the murine cough model or secretion experiment, displayed both expectorant and antitussive effects, respectively (*Kuang et al., 2018*). In the

text below, the data on expectorant or antitussive effects of the herbal drugs or their pure constituents are provided. For each effect, a brief background information is given to clarify the possible mechanism of action in a comprehensive way. Results from the pharmacological *in vivo* and *in vitro* research are summarized and critically discussed.

# REPORTED EFFECTS OF COUGH PHYTOTHERAPY

## Expectorant effects

The bronchial epithelium is predominantly formed by ciliated cells and interspersed with mucus-secreting cells. The mucus-secreting cells include the mucous, serous, Club (previously known as Clara) and dense-core granulated (DSG, syn. Kulchitsky) cells of the surface epithelium, and the mucous and serous acinar cells of the submucosal glands. The mucous cells produce acidic high molecular weight glycoprotein, while the serous cells serve as the source of neutral mucin and the Club cells of carbohydrate components of surfactants and antiproteases. The DSG cells are pulmonary neuroendocrine cells. The number of mucus-secreting cells can increase under non-physiological conditions, *e.g.*, chronic bronchitis or following the inhalation of irritants (*Jeffery & Li, 1997*).

Normal human airways are covered by mucus which secures a variety of protective functions, including the entrapment and clearance of exogenous materials by ciliary transport and the moistening of the epithelial surface. Roughly 100 mL of this fluid is physiologically produced every day, and its presence in the airways does not cause any irritation. The mucus layer is mainly composed of water, glycoproteins, glycosaminoglycans, proteins and peptides, lipids, antiproteases, antioxidants and ions of both epithelial and vascular origin. The airway mucin glycoproteins (mucins) are high molecular weight glycoconjugates which are linked by disulfide bridges and make the mucus viscous. The mucins significantly contribute to normal mucociliary clearance and innate immune system in the airways. The viscosity of the mucus also depends on ion content. For example, the amount of calcium and sulfate in tracheobronchial secretion is increased in cystic fibrosis (*Boat et al., 1976*).

Under pathological conditions, more mucus with higher viscosity is produced. Acute challenge of the airways by environmental irritants, allergens, or infectious pathogens activates inflammatory/immune mediators, and some of these activate both the mucus-secreting cells on the epithelial surface and the submucosal glands with triggering of the mucin hypersecretion. Mucin overproduction is a typical feature of chronic respiratory diseases, such as asthma, COPD and cystic fibrosis. In severe cases, the airways can almost completely be occluded. Enhanced mucus viscosity is the result of both higher concentration of the mucins and the amount of the disulfide links among them. The composition of the mucus can also be varied, for example, the mucin MUC5B: MUC5AC ratio is increased in chronic bronchitis sputum (*Rose & Voynow, 2006*).

Excessive mucus is coughed out of the airways. This productive cough can be facilitated by expectorants, which act in three principal ways. (1) Secretolytic agents stimulate the submucosal glands to secretion of watery mucus, which is easier to expectorate. Saponins can serve as a typical example of natural secretolytic agents. The substances are both water

and fat soluble, and act as surface-active agents (surfactants). After oral administration, saponins irritate the stomach mucous membrane and trigger a reflex response with the enhancement of respiratory secretion through an afferent stimulus. In high doses, their administration can result even in vomiting due to the vomiting center stimulation (*Petrovic et al., 2022*). (2) Secretomotoric agents activate cilia and increase the efficiency of the mucociliary clearance in this manner; the mucus is then removed more efficiently. Among natural substances, this effect is exerted *e.g.*, by essential oils. (3) Mucolytic agents make the mucus less viscous by the depolymerization of the glycoprotein molecules, and facilitate expectoration. An expectorant substance can act through more than one of these mechanisms.

### Experimental methods

The expectorant effects of the herbal drugs and their active constituents were mostly studied using cell cultures, where the stimulation of the secretion or stimulation of the cilia was investigated.

### Pharmacological studies on the expectorant effects

#### In vivo

Expectorant effects *in vivo* were confirmed in some *Glycyrrhiza* spp. components for an ointment containing menthol and camphene (*Kuang et al., 2018*; *Schäfer & Schäfer, 1981*; *Schönknecht et al., 2017*). In the murine model based on the determination of the phenol red secretion, the *Glycyrrhiza* spp. components (liquiritin apioside, liquiritin, or liquiritigenin) exerted expectorant activities after a 3 day treatment (50 mg/kg *per os*). Surprisingly, potent antitussive activities of the same substances were shown as well (see below) (*Kuang et al., 2018*). Bronchodilatory and secretolytic effects of the insufflated ointment on the basis of menthol, camphene and essential oils (Pinimenthol®) were observed in the guinea pig model of acetylcholine induced bronchospasms (50% bronchospasm reduction and 44% increase in tracheobronchial secretion). Surprisingly, a higher secretion increase (54%) was found after the application of the ointment on the epilated skin (*Schäfer & Schäfer, 1981*).

#### In vitro

Increased secretion of watery mucus by bronchial cells (*i.e.*, secretolytic effects) after administration *Hedera* saponins ($\alpha$- or $\beta$-hederin, respectively) or *Glycyrrhiza* spp. (glycyrrhizin) is explained by an indirect $\beta_2$-mimetic mode of action. In the case of $\alpha$-hederin, inhibition of $\beta_2$-receptors internalization and increase in $\beta_2$-receptor binding were observed. These processes result in an increased intracellular cAMP level, which subsequently elevates surfactant production, leading to a decreased mucus viscosity, and secretion in the alveolar type II cells. This may explain the secretolytic and bronchospasmolytic effect of the aforementioned saponins (*Greunke et al., 2015*; *Sieben et al., 2009*).

Menthol stimulated $Cl^-$ secretion across canine tracheal epithelium, probably through a $Ca^{2+}$-dependent mechanism. The monoterpene thus may influence mucociliary transport in the respiratory tract (*Chiyotani et al., 1994*). Reduced secretion of certain mucins

(mainly MUC5AC, MUC2, MUC19) was demonstrated, for example, for 1,8-cineole (*Sudhoff et al., 2015*), glycyrrhizin (*Ho-Jin et al., 2006*; *Li et al., 2018*; *Nishimoto et al., 2010*) and for the fixed combination of the Thymi herba and Primulae radix extracts (*Seibel et al., 2018a*) in various cell cultures. The reduction in mucin production was dose-dependent, with the effective concentrations in *in vitro* experiments being roughly in tens to hundreds of μM. The mechanism of action is not fully clear, attenuation of the transcription factor nuclear factor-κB (NF-κB) activity was suggested (*Sudhoff et al., 2015*). The secretomotoric effect is addressed less often, but is not irrelevant. Glycyrrhetinic acid promoted the differentiation of the nasal epithelial cells by the stimulation of the re-epithelialization of the nasal mucosa and increased the number of cell cilia in the patients suffering from chronic vasomotor rhinitis (*Passali et al., 2017*). There is also evidence of a secretomotoric effect for certain herbal drugs such as Foeniculi amari fructus, Foeniculi dulcis fructus and Anisi fructus. Notably, some other herbal drugs used in cough therapy had no effect on the mucociliar transport rate (Plantaginis lanceolatae folium and Liquiritiae radix) or caused its inhibition (Althaeae radix) (*Müller-Limmroth & Fröhlich, 1980*). The effects of *Thymus* spp. are diverse (*Müller-Limmroth & Fröhlich, 1980*; *Nabissi et al., 2018*; *Wienkötter et al., 2007*). *Thymus* extract is a complex mixture of essential oil (mainly thymol and carvacrol) and flavonoids (apigenin, luteolin and their glycosides). Bronchospasmolytic effects of thymol are related to its agonism on $\beta_2$-receptors (*Beer, Lukanov & Sagorchev, 2007*), to the blockade of $Na^+$ channels (*Haeseler et al., 2002*) and the activation of the TRPA1 channels (at small thymol doses) (*Lee et al., 2008*). This latter effect is also exhibited by carvacrol (*Xu et al., 2006*). Additionally, apigenin, luteolin and its glycosides were identified as possible spasmolytic compounds (*Van Den Broucke & Lemli, 1983*). Last but not least, Thymi herba extract inhibited *ex vivo* endothelin-induced contraction of isolated rat trachea smooth muscles *via* non-competitive interaction with endothelin receptors. However, thymol was found to be an inactive extract component (*Engelbertz et al., 2008*).

Notably, stimulation of respiratory bitter taste receptors (TAS2R) also leads to bronchodilation (*Liggett, 2013*). Since various natural compounds are TAS2R agonists (*Meyerhof et al., 2010*), this effect could represent another mechanism of their action. Also noteworthy, the essential oils produce other important effects, including anti-inflammatory and antibacterial (see below).

*Clinical evidence*

1,8-Cineole (eucalyptol), used to relieve nasal congestion in the patients with common cold, is the principal constituent of the essential oil of eucalyptus. The oil probably exhibits expectorant and mucolytic effects (*Dixit, Rohilla & Singh, 2012*) in the bronchial epithelium. Administration of 1,8-cineole to asthmatic patients resulted in a substantially improved lung function and alleviation of dyspnoea (*Worth & Dethlefsen, 2012*). 1,8-cineole also alleviated the discomfort of non-purulent rhinosinusitis in the patients with acute rhinosinusitis (*Kehrl, Sonnemann & Dethlefsen, 2004*).

Interesting clinical study (randomized, double-blind controlled) dealt with the effect of a spray containing the essential oils of *Eucalyptus citriodora*, *E. globulus*, *Mentha × piperita*, *Origanum syriacum*, *Rosmarinus officinalis*, and a placebo spray (application five

times a day over 3 days). Compared to those from the placebo group ($P = 0.019$), the members of the study group felt a more pronounced relief from the symptoms that included a sore throat, hoarseness, or cough, after a 20 min period. In 3 days, however, the differences in the intensity of the symptoms were negligible ($P = 0.042$) (*Ben-Arye et al., 2011*).

As regards the efficiency of *Hedera helix* leaf extracts as an expectorant, the clinical data available are not conclusive. According to a review summarizing data from 13 clinical trials, herbal preparations from *H. helix* could serve as a remedy for early symptoms of respiratory tract infections, regardless of age. Sufficient clinical evidence has been gathered to prove the positive effects of ivy preparations in cough therapy. However, the amount of data on an improvement of cough expectoration, night coughing and cough-associated sleep disruption is insufficient (*Barnes et al., 2020*). Interestingly, yet another study reported that Hederae folium extract could serve as an alternative to acetylcysteine in the improvement of the respiratory function in both children and adults. In both groups, alleviation of all symptoms was recorded after 7 days. Notably, a more pronounced improvement in the number of cough attacks and cough-related sleeping disorders was observed in the participants taking the *Hedera* syrup (*Kruttschnitt et al., 2020*). Effectiveness and safety of dry *Hedera helix* L. extract was also studied in a non-randomized, non-interventional, multicenter, open-label, post-authorization effectiveness study (464 patients aged 2–12 years with productive cough). Significant improvement in several parameters (cough, chest pain, wheezing, dyspnoea, auscultation changes or body temperature) and no adverse drug reactions were reported (*Schönknecht et al., 2017*). Yet another systematic review, focused on the use of Hederae folium extracts for the treatment of acute upper respiratory tract infections, confirmed their safety as cough medications. By contrast, as regards minimizing cough severity and/or frequency, the clinical significance of the data seemed very poor (*Sierocinski, Holzinger & Chenot, 2021*). Finally, a retrospective, multicenter cohort study of the expectorant properties of the *Hedera helix* leaf extracts in pregnant women revealed their safety for the fetus (*Alkattan et al., 2021*).

As regards the preparations from thyme, children 2 months to 14 years old suffering from a bronchial catarrh or bronchitis were given a dose of a thyme syrup over the period of 7–14 days in an open study. Alleviation of coughing was recorded in 94% of the patients (*European Scientific Cooperative on Phytotherapy, 2003*). By contrast, another study (randomized, double-blind) dealt with patients with a productive cough, accompanying uncomplicated respiratory infections. A thyme syrup or a bromhexine preparation was administered to the patients for 5 days, but no appreciable difference between the two interventions was reported (*Knols, Stal & Van Ree, 1994*). While these studies are rather unconvincing in terms of clinical efficacy, long-term use of *Thymus* spp. is supportive of the application of various products from this plant, including the plant material, extracts and the essential oil as traditional herbal remedies for the cold-accompanying productive cough. Thyme components were also employed as components of fixed combination with those from other plant species. Specifically, a fixed combination of Thymi herba and Primulae radix extract reduced the Bronchitis Severity Score (BSS) in acute bronchitis,

both in adults and children (*Gruenwald, Graubaum & Busch, 2005*, *2006*; *Kemmerich, 2007*; *Ludwig, Stier & Weykam, 2016*).

Next, just a single study on licorice revealed that licorice pastille significantly decreased the Leicester Cough Questionnaire scores of the patients with a chronic cough (*Ghaemi et al., 2020*).

More data are available on fennel. Fennel seeds facilitate the transport of extraneous particles *via* the enhancement of the ciliary motility of the respiratory apparatus. The inhalation of fennel essential oil stimulates the smooth muscle contraction of the trachea and, consequently, a smoother expectoration of the mucus, bacteria, and other foreign corpuscles occurs. *trans*-Anethole, the principal constituent of the oil, is also analogous to catecholamines, namely epinephrine, norepinephrine, and dopamine. It is therefore not surprising that various sympathomimetic effects of fennel, including a bronchodilatory activity, were observed (*Javed et al., 2020*). The efficacy of *Zataria multiflora* and *Foeniculum vulgare* essential oils (Tussivin® oral drops) in comparison to clobutinol and placebo in the treatment of acute cough in 119 patients was reported in a clinical study. The dose of 20 drops of each preparation three times a day over 3 days was administered to the patients, aged over 14. Undisputable recovery in the patients suffering from a long-term cough, lasting for more than 7 days, was observed when Tussivin® oral drops were used (*Afzali, Kashanian & Akbari, 1998*; *Mahboubi, 2018*).

Finally, studies of the effect of aniseed (Anisi fructus, *Pimpinella anisum*) in humans are rare. The effect of the water extract from the seeds was examined in 26 patients with a chronic rhinosinusitis with no polyps (200 μg/nostril *vs* one fluticasone dose/nostril; in 12 h intervals for 4 weeks). Both preparations alleviated the symptoms of mucosa inflammation and sinonasal disorders. In comparison to fluticasone, anise caused a more pronounced improvement of the scores for both the sinonasal outcomes and rhinological symptoms (*Singletary, 2022*).

## Discussion

The mechanism of the expectorant effect is probably not the same for all herbal drugs or their active constituents. The secretolytic and secretomotoric effects are among the most important ones. Out of the active principles, saponins and essential oils are likely to play the principal role. Basic research results imply that these agents mainly trigger the secretion of more watery mucus and reduce mucine production along with a change in mucine spectrum. These principal effects can be supplemented by others. The activation of $\beta_2$ adrenoreceptors in the respiratory system leads to bronchodilation. For some agents, this effect may be related to certain chemical similarity with catecholamines. The TAS2R receptors represent yet another receptor type, whose activation could result in bronchodilatory effects. In addition, many types of membrane channels ($Na^+$, TRPA1, $Cl^-$) can be targeted by expectorants.

Some clinical studies confirm positive effects on expectoration in various subjects (asthmatics, patients with rhinosinusitis or respiratory infections, adults *vs* children, nocturnal coughing), other studies are less convincing. For the conclusions obtained, the study design and the method of evaluation are crucial. The latter may be questionable,

because the efficacy of the therapy is often assessed by questionnaire scores, where the results obtained are largely subjective.

## Antitussive effects

Suppression of severe and persistent cough is highly desirable. At the very beginning, dry cough should be very carefully distinguished from productive cough. Caution is advised principally in the patients with impaired coughing (such as the patients with COPD at stage IV). Notably, dry cough can indicate serious diseases, including cancer. In such cases, cough suppression brings relief, but does not influence the original pathology. Conventional medicines are classified as peripheral or central antitussives. The peripheral antitussives such as butamirate display peripheral local anesthetic action on the afferent sensory nerves in the trachea and larynx. The conduction of the irritating stimuli is thus reduced, and the reflex response prevented. The central antitussives, such as codeine, suppress the cough center in the CNS that controls cough by agonism on the opioid receptors.

Although herbal remedies have been used to alleviate the dry cough since ancient times, the underlying mechanisms of their action remain not fully understood. The suppression of cough is principally attributed to a decrease in the local irritation which is mainly ensured by mucilage composed of high number of monosaccharides linked by glycosidic bonds. These polymers can be linear or branched, and their molecules can be constructed from either identical monosaccharides (homopolysaccharides) or different monosaccharides (heteropolysaccharides). In general, branched molecules or presence of charged groups (sulfate, carboxylate) ensure a good water solubility. The solubility is rather poor in the case of linear chains or large molecular weight polysaccharides. Importantly, some polysaccharides gelatinate under certain circumstances. Thus, their interaction with water on the airway surface results in the formation of a thin protective layer. In addition to protection from local irritation, this polysaccharide layer also constitutes a remedy for tissue rehydration and improved shielding against physical or microbial stress impulses (*Sendker et al., 2017*). Accordingly, the herbal drugs traditionally used for cough suppression usually possess a high content of polysaccharides, for example, Althaeae radix up to 11.6%, Lichen islandicus up to 50%, and Plantaginis lanceolatae folium approx. 6% (*Nagy, Mučaji & Grančai, 2017*). Surprisingly, the dry root extract from *Pelargonium sidoides* (EPs® 7630) contains mainly monomeric and oligomeric saccharides (*Schoetz et al., 2008*).

### Experimental methods

Conclusive research on cough suppression is not easy. To confirm a real antitussive action, the use of *in vivo* cough models is indispensable. Although there is a strong tendency to reduce the usage of laboratory animals in accordance with the three Rs rule (Replacement, Reduction and Refinement), animal experiments undoubtedly provide the best evidence of antitussive effects. Various *in vivo* models of cough differing in species applied and in cough induction have been developed (*Bolser, 2004*; *Lewis et al., 2007*). As recently shown in an immunohistochemical study, the morphology of human lungs is most similar to that

of guinea pigs (*West et al., 2015*). Accordingly, the guinea pigs are probably the most frequently used, being most suitable for *in vivo* cough experiments, followed by cats or, less often, by mice, rats, dogs and pigs. In general, the animals are naïve or pre-treated (allergized, exposed to cigarette smoke *etc*). Cough can be induced by a protussive agent, for example capsaicin, citric acid, ammonia or an antigen to which the animal has been sensitized, usually administered in the form of an aerosol. The reflex can also be provoked mechanically, for example with a thin synthetic fiber (*Nosáľvá et al., 1992*). In line with the real way of administration in humans, that of the tested substance or extract is often oral. As a reference drug, codeine is considered a classical standard for the antitussive effect evaluation.

With some limitations, a decrease in the local irritation of the airways by the herbal polysaccharides can be studied *in vitro*. Of these experiments, the tests of the bioadhesion of plant mucilage to the airway epithelial mucosa are the best developed. For this purpose, systems based on the porcine colonic tissue or porcine buccal membranes have been employed. A direct histological proof of the polysaccharide adhesion using the fluorescent-labeled rhamnogalacturonans and fluorescent microscopy is another useful option. Both models have brought interesting results: (1) the polysaccharide layer is located exclusively on the apical membrane, while there is no adhesion to the serosal membrane (results obtained on the colonic tissue). (2) The layer has a diffuse character, and the binding of the polysaccharides to specific membranal receptors is thus unlikely. The polysaccharide-polysaccharide interaction with the endogenous mucins was hypothesized, namely between the glycosylated regions of the mucins and the galacturonide regions of the rhamnogalacturonans. (3) There is a relationship between the chemical structure of a given polysaccharide and its adhesivity. Acidity of the molecule is important, since bioadhesivity was found for acidic polysaccharides, but not for neutral ones (arabinogalactans, dextrans and xyloglucans). Highly esterified and thus neutral molecules were also ineffective, which again indicates the role of acidity and polar properties. Linearity of a molecule supports the process, while branching or non-linear backbone structures reduce bioadhesivity. Hence, the linear, strongly acidic homogalacturonans are the best bioadhesive agents reported at least in these experiments. A significant bioadhesion was also observed for rhamnogalacturonans with a low degree of esterification, and for linear oligogalacturonans derived from pectin (rhamnogalacturonans are among the acid soluble plant polysaccharides) (4). The bioadhesive effects are concentration dependent.

### Pharmacological studies on the antitussive effects

#### In vivo

Among the antitussive medicinal plants studied with the *in vivo* cough models, *Althaea officinalis* is the most frequently reported. Its cough suppressing action could be, in principle, based on the mucilaginous polysaccharides, mainly rhamnogalacturonans (*Nosáľvá et al., 1992*, *1993*; *Šutovská et al., 2011*, *2007*, *2009*). The antitussive effects were also documented for *Plantago lanceolata* and *Pelargonium sidoides* in the form of extracts (*Bao et al., 2015*; *Boskabady et al., 2006*). Certain effects were reported for some coumarins

and flavonoids (*Huang et al., 2020*). Limited amount of data on the decrease in cilia beating and mucus clearance are available (inhibition for Althaeae radix (*Müller-Limmroth & Fröhlich, 1980*) and stimulation for *Pelargonium sidoides* root extract (*Neugebauer et al., 2005*)). Surprisingly, significant antitussive effects have been described also for *Glycyrrhiza* spp. constituents (liquiritin apioside, liquiritin, and liquiritigenin at 50 mg/kg *per os*) in the murine model of cough, induced by ammonia. These effects were partially antagonized by pretreatment with a 5-HT receptor antagonist (methysergide) or with a $K_{ATP}$ channel blocker (glibenclamide), which implies the participation of these structures (*Kuang et al., 2018*).

*In vitro*

In the *in vitro* studies, a moderate bioadhesion to the porcine buccal epithelial tissue has been confirmed for the mucilaginous polysaccharides from *Althaea officinalis* and *Plantago lanceolata*, and the effects were concentration-dependent in both cases (*Schmidgall & Hensel, 2002*; *Schmidgall, Schnetz & Hensel, 2000*). Importantly, there are differences among the cell lines. The polysaccharides were actively taken up by the epithelial cells and could thus exhibit intracellular effects, as found for those from *Althaea officinalis*. By contrast, fibroblasts in the same experiment simply absorbed the polysaccharides on their surface, and no cellular internalization of the polymers was observed (*Deters et al., 2010*).

*Clinical evidence*

The only clinical studies that involved syrup preparations combining more herbal components (*Althaea officinalis*, *Cetraria islandica*, *Plantago lanceolata* and *Pelargonium sidoides*) were published recently. Therefore, a synergistic action of individual components should be anticipated. Relevant molecules in the mixtures (*e.g.*, mucilage, iridoids, flavonoids, phenylethanoids, phenolic acids) act in complexes, exhibiting their effects through diverse mechanisms. While each individual component contributes to clinical evidence, the result is not just a simple sum of individual effects. No product tested was a traditional herbal medicinal product. Thus, the preparations were planned to be marketed under the medical device legislation (as opposed to the European medicinal products for human use legislation), which also significantly influenced the conclusions of these studies.

The following plants are separately used in various preparations in European official phytotherapy: *Althaea officinalis* (*EMA, 2016*), *Cetraria islandica* (*EMA, 2014a*), *Plantago lanceolata* (*EMA, 2014b*) and *Pelargonium sidoides* (*EMA, 2018a*), and categorized as "traditional use". With the exception of *Pelargonium sidoides*, no reliable clinical studies have been performed for the others, with the registration of the corresponding single-component herbal products being allowed by the pertinent legislation.

Besides the clinical trials published up to and including 2018, summarized in the Assessment Report (*EMA, 2018b*), some more recent data for *Pelargonium sidoides* are available. *Martin et al. (2020)* found out that the extract from *Pelargonium sidoides* root helped to significantly reduce the need for concomitant antibiotic prescriptions, and substantially shortened the time of patient recovery. The results were published in a

retrospective cohort study in 117,182 patients 20–60 years of age. In another interesting study, the effect of the extract from the roots of *P. sidoides* on the immune response of athletes was investigated by *Luna et al. (2011)*. The extract enhanced the production of immunoglobulin A in the saliva and reduced the levels of interleukin (IL) IL-15 in the nasal mucosa and that of IL-16 in the serum (*Luna et al., 2011*). *Riley et al. (2019)*, administered 40 mg EPs[®] 7630 or placebo three times a day for 10 consecutive days to a set of 105 patients (men and women 18 to 55 years of age) with common cold symptoms (nasal drainage, sneezing, sore throat and nasal congestion, hoarseness, muscle aches, headache, scratchy throat, cough, or fever). Compared to the placebo group, EPs[®] 7630 alleviated cold symptoms in the other group (*Riley et al., 2019*). Similarly, the effects of *P. sidoides* extract EPs[®] 7630 in the treatment of acute respiratory tract infections (RTI) in children was reviewed by *Careddu & Pettenazzo (2018)*. A significant improvement in the intensity of RTI symptoms, excellent safety and tolerability of EPs[®] 7630 was evidenced by all clinical studies (*Careddu & Pettenazzo, 2018*). Notably, the potency of EPs[®] 7630 in treating human coronavirus infections (HCoV) was also investigated. A set of 61 patients, out of which 37.7% were infected with a HCoV strain, and had the symptoms of common cold with an accompanying viral infection, were given 20 mg of EPs[®] 7630 three times a day. A significant improvement of the cold symptoms, especially in the HCoV group, was observed. A complete recovery or significant alleviation of the symptoms was recorded in 80% of the patients. In addition, the majority of the patients reported satisfaction with EPs[®] 7630 treatment (*Keck et al., 2021*). A recent meta-analysis summarized the data from seven clinical trials with a total of 1,099 participants. While adult patients had the symptoms of common cold, the child group also included those suffering from an acute non-streptococcal tonsillopharyngitis. Both groups reported a sore throat and hoarseness. Again, alleviation of both symptoms and faster recovery in the EPs[®] 7630 treated groups *vs* the placebo was reported (*Kamin et al., 2022*). Finally, according to *Seifert et al. (2019)*, the use of EPs[®] 7630 reduced the symptoms, decreased the need for parallel administration of paracetamol, and helped to achieve a faster recovery, based on reviewing six clinical trials with 523 children aged 6 to 10 years with acute bronchitis or non-streptococcal tonsillopharyngitis.

### Discussion

Regardless of the cough model used, the *in vivo* effective antitussive doses were roughly 50 mg/kg orally. The effects were dose-dependent, and not long lasting which implies the need for repeated administration. The *in vivo* toxicity of the herbal therapy was apparently low in all experiments. Notably, the cough reflex threshold may be decreased by some expectorant constituents, as reported for menthol (the principal component of menthae piperitae aetheroleum) (*Wise, Breslin & Dalton, 2012*).

### Anti-inflammatory effects

Inflammation, a key defense mechanism, is a reaction triggered by damage to living tissues. Airway inflammation is one of the most common causes of cough, both productive and dry. The inflammatory response is a very complex process, and many factors are involved

in its initiation, progress, and termination. The following text will only highlight the main players and principles, which are important in the cough pathology and therapy. The airways are continuously exposed to the air which contains potentially harmful components, including gases, particles, bacteria, viral particles and fungi. Although their penetration is size- and shape-limited and only those smaller than 5 μm can get deeply into the lungs, continuous defense of the airways is necessary. Immunologically, there are three classical defense lines: a physical and chemical barrier, a non-specific innate response, and a specific adaptive response. (1) The physical and chemical barrier on the airway surface is provided by the mucociliary system which has already been mentioned. Physiologically, the cilia beat more than 1,000 times a minute and move the mucus upwards about 0.5–1 cm per minute. The sticky mucus layer traps the particles (including pathogens) and the cilia move the mucus up where it is coughed out or swallowed. If the pathogens penetrate the protective mucosa, (2) the non-specific innate response that serves as the second line of defense comes into play. The macrophages on the alveolar wall or free in the lumen continuously sample the airspace, ingest the deposited particles and kill the living ones in the lysosomes. After the destruction of the pathogen, the antigenic fragments are displayed on the cell surface and recognized by the cytotoxic T cells which subsequently destroy the attacked cell. The macrophages produce both pro-inflammatory and anti-inflammatory cytokines, chemokines and growth factors, which coordinate the response. If necessary, humoral innate immunity, based on soluble factors, also participates. Among the factors, complement is the best known, followed by pentraxins and collectins. These biomacromolecules are evolutionarily highly conserved proteins, which recognize and bind pathogens and endogenous ligands. The recognition of the pathogens is mediated by the pathogen-associated molecular patterns (PAMPs) such as lipopolysaccharide (LPS) of Gram-negative bacteria, lipoteichoic acid (LTA) of Gram-positive bacteria or β-glucans of fungi. The soluble factors can also be forced to act by endogenous damage/danger-associated molecular patterns (DAMPs) (*Kovach & Standiford, 2011*), such as hyaluronic acid or heat-shock proteins. The complement can be activated by the coagulation cascade (*Hamad et al., 2012*). Upon activation, the humoral immunity proteins recruit additional leucocytes from circulation to support the fight against the noxious stimuli. Among them, mainly neutrophils are endowed with a number of mechanisms for pathogen elimination, including phagocytosis, generation of reactive oxygen species (ROS) or the formation of neutrophil extracellular traps. Additional macrophages derived from monocytes can be called up as well. Accumulation of the white blood cells at the site of infection results in inflammation. By contrast, (3) the specific adaptive response, based on B and T lymphocytes, is not site-limited, but occurs throughout the body. The B lymphocytes engulf the "known" antigen and display the antigenic fragments bound to major histocompatibility complex (MHC) on their surface. As a consequence, the helper T lymphocytes secrete interleukins, which convert the B lymphocytes into antibody-producing plasma cells. Additionally, the T lymphocytes can be activated by dendritic cells, which present antigens as well. Some of the B cells are
transformed into memory cells, capable of activation upon future contact with the pathogen. The plasma cells secrete antibodies which have various important roles, such as the formation of complexes with antigens thereby preventing their binding to the host cells, and marking the pathogens for the phagocytes. The antigen-antibody complexes activate the complement proteins, which consequently kill the pathogens by rupturing their cell membranes. The complement proteins further increase the accumulation of the leucocytes at the infection site and promote inflammation. The activated T cells secrete the cytokines which trigger the maturation of the T cells into various subtypes. The killer T cells are cytotoxic and destruct the infected cells. The helper T cells assist the killer T-cells and the B lymphocytes in their functions. The regulatory T cells suppress other T lymphocytes and prevent the exaggerated immune response. Importantly, some T cells mature into memory T cells that initiate an immune response if the body encounters the same pathogen again (*Sanchari Sinha Dutta, 2021*). The role of the airway sensory nerves cannot be omitted when addressing the inflammation-induced cough. Upon activation, these neurons not only conduct the signals to the brain with triggering of the cough reflex, but also release neuromediators, which locally modify the inflammation.

The airway epithelium is not a simple barrier and producer of mucins. This tissue also possesses sensoric properties, and very closely collaborates with the immune and neural cells. The airway surface accommodates pattern-recognition receptors, such as the Toll-like receptors (TLRs) which recognize microbial noxious substances and participate in the initiation of the immune response (*Arora et al., 2019*). To date, eleven TLRs have been identified in humans. Some of them are located on the cell surface and principally serve for the recognition of bacterial structures, others are endosomal and recognize nucleic acids. When activated, the TLRs can activate the myeloid differentiation gene 88 or spark interferon signaling. Similarly, the cytokine receptors sense the signals from the immune cells, and mediate a cross-talk among epithelial cells, alveolar macrophages, dendritic cells and the memory T-cells. Tight junctions among adjacent epithelial cells secure not only the barrier integrity, but also the separation of apical and basolateral sides of the epithelial cells and the corresponding receptors. The barrier integrity and receptor separation are disturbed after epithelium disruption by an injury. Moreover, advanced sequencing techniques helped to identify three novel epithelial cell types which may play important immune roles. (A) The tuft cells are hypothesized to have chemo-sensorial properties and to be activated with the participation of the TAS2R receptors, $M_3$ receptors and TRPM5 channels. Once activated, the cells may release mediators, such as acetylcholine, eicosanoids and cytokines. The tuft cells are localized in the proximity of the nerve fibers, and probably mediate the communication between the neuronal and immune pathways. They are known to promote the protective respiratory reflexes, such as sneezing, as well as the local neurogenic inflammation of the respiratory mucosa. (B) The pulmonary neuroendocrine cells (PNECs) are the only innervated airway epithelial cells. They respond to the changes in oxygen levels, stretch and chemical stimuli by the production of neuropeptides and neurotransmitters. The PNECs are assumed to participate in the

coordination of the immune response and to be of particular importance in early life. (C) The ionocytes significantly express the CFTR gene that encodes the critical chloride-ion transporter. Its deficiency leads to increased mucus viscosity, impaired mucociliary clearance and chronic airway inflammation and infection (*Hewitt & Lloyd, 2021*). Notably, cystic fibrosis is associated with the mutations in the CFTR gene (*Farinha & Callebaut, 2022*). Surprisingly, the airway epithelium also participates in apoptosis. Analogously to macrophages and dendritic cells, the epithelial cells are capable of sensing and engulfing the apoptotic cells. This ability is important, since it prevents a significant inflammatory response of the airways to common inhaled allergens. However, in the long-term, hyperplasia of the basal epithelial cells which represents the first step in the development of COPD, an inflammatory airway disease accompanied by a typical cough, can arise. Other roles of the airway epithelium are also likely. The basal epithelial cells may have an intrinsic capacity for inflammatory memory, as uncovered for chronic rhinosinusitis with polyps (*Hewitt & Lloyd, 2021*; *Ordovas-Montanes et al., 2020*). The cells also participate in the local circadian control, including the time-of-day variation in the inflammatory response (*Hewitt & Lloyd, 2021*; *Gibbs et al., 2009*). This phenomenon could play a role in the night attacks of cough or in the progression of asthma, and deserves further attention.

## *Experimental methods*

As the airway inflammation represents an important triggering factor of both the productive and dry coughs, the general anti-inflammatory effects of the cough phytotherapy, both expectorant and antitussive, were investigated in many studies. Although the use of *in vitro* methods dominates, *in vivo* studies on various animal models of inflammation have also been published. The model of acute lung injury (ALI), induced by the intratracheal administration of LPS is the most popular, followed by the models of lung infection, allergic asthma, and COPD. The models of inflammation in other tissues, such as the paw edema or various arthritis models, are also relatively frequently employed. The evaluation of the anti-inflammatory effects varies from case to case and includes, for example, the determination of the histological changes, the bronchoalveolar lavage fluid (BALF) analysis of protein levels and infiltration by the inflammatory cells or measurement of lung functions by whole-body plethysmography. Outside the respiratory system, parameters such as the edema or pain reductions are determined. Both plant extracts and pure substances were tested. Administration was usually oral, both at single and repeated doses. In case repeated administration was used, its duration was most often in the range of days up to 1 month. Injection (i.p., i.m.) or inhalation are also frequent.

*In vitro* studies on the anti-inflammatory effects with extremely variable designs were published. The data were mostly obtained on immune cells, typically on the macrophages or other granulocytes, and on the dendritic cells. Other cell lines were employed less frequently, and only some of them are directly related to the respiratory system, such as the airway endothelial cells. Somewhat surprisingly, non-respiratory cell lines, such as the human umbilical vein endothelial cells, were used in the cough phytotherapy research, too. Specific processes related to the immune responses were often studied, including phagocytosis, the cell migration or the effects on the complement.

### Pharmacological studies on the anti-inflammatory effects

*In vivo*

Of the expectorant plants studied *in vivo*, the *Hedera helix* in the form of extract or its fraction has probably been the most frequently investigated. Its anti-inflammatory effects were shown in the ALI model (*Shokry et al., 2022a*) as well as in the model of arthritis (*Shokry et al., 2022b*). The effective doses were roughly 100–200 mg/kg. The same effects were shown for its principal component, hederacoside C (5–50 mg/kg i.v. or i.p.) (*Akhtar et al., 2019*; *Han et al., 2022*). *trans*-Anethole, the principal component of Anisi aetheroleum or Foeniculi amari fructus aetheroleum, also exhibited anti-inflammatory action in the ALI and COPD models in the effective doses ranging from 62.5 to 500 mg/kg i.p. or orally (*Kim, Lee & Seol, 2017*). Besides the respiratory system, *trans*-anethole was effective in the paw edema, pleurisy or ear edema models of inflammation in similar dose ranges (*Domiciano et al., 2013*; *Wisniewski-Rebecca et al., 2015*). 1,8-Cineole, the principal component of Eucalypti aetheroleum, ameliorated inflammatory changes in three different models of the lung injury–the ALI, allergic asthma and COPD. Inhibition of the inflammatory cell infiltration was observed in all cases. The treatment was oral (up to 260 mg/kg) or by inhalation (10 mg/mL for 30 min, 6 h before the allergic challenge). In the case of the COPD model, the administration was repeated over 12 weeks (*Lee et al., 2016*; *Yu et al., 2018*; *Zhao et al., 2014*). Importantly, in both long-term respiratory pathologies (asthma and COPD), restoration of the impaired lung functions was confirmed (*Lee et al., 2016*; *Yu et al., 2018*). Anti-inflammatory properties were also reported for the licorice root extract and its main components, glycyrrhizin and liquiritigenin. Attenuated mucus hyperproduction and goblet cell hyperplasia were shown after glycyrrhizin subcutaneous administration in two different murine models of lung inflammation (intratracheal instillation of LPS or of IL-4). These effects were at least partially mediated by the inhibition of the MUC5AC gene transcription (*Nishimoto et al., 2010*). A mixture of *Thymus* and *Primula* extracts had similar effects in a rat model of inflammation in tens to roughly 400 mg/kg given orally in one single dose or three consecutive doses (*Seibel et al., 2018a*, *2018b*). Apart from the respiratory system, the licorice root extract and liquiritigenin had positive effects on the murine liver inflammation (*Yu et al., 2015*) and rat paw edema (*Kim et al., 2008*), respectively. The doses of liquiritigenin were 15 mg/kg given intravenously for 2 days or 50 mg/kg orally for 3 days, and that of the extract 300 mg/kg orally for 3 days.

Among the antitussive plants, anti-inflammatory *in vivo* effects on the airways were reported for the extracts from *Pelargonium sidoides* (EPs® 7630) and from *Althaea officinalis*. In both cases, the effective doses were roughly 50–100 mg/kg. Positive changes were observed in the tracheal tissue and were dose-dependent, even though the way of administration was different (orally or by inhalation) (*Bao et al., 2015*; *Kolodziej, 2011*; *Mikaili et al., 2010*). As regards pure substances, scopoletin and plantamajoside showed anti-inflammatory properties in murine models of lung injury (both administered i.p. at the doses between 10–100 mg/kg) (*Leema & Tamizhselvi, 2018*; *Wu et al., 2016*). Besides the airways, positive effects on wound healing and pain were observed together with an

increase in the phagocytic activity of the macrophages or decrease in the leukocyte infiltration. In these aspects, the studies mostly showed the effect of the *Althaea officinalis* extract. Some data on the extracts from *Cetraria islandica* and *Plantago* spp. are also available. Both the ways of administration (orally, i.p., s.c. or topical) and the effective doses (from 2.5 to 200 mg/kg) in these experiments were variable (*Freysdottir et al., 2008*; *Marchesan et al., 1998*; *Rezaei et al., 2015*; *Shipochliev, Dimitrov & Aleksandrova, 1981*; *Valizadeh et al., 2015*; *Yazdian & Irani, 2015*). From pure substances, anti-inflammatory effects in other tissues were reported for scopoletin (5 and 10 mg/kg, i.p.) in the murine model of the paw edema (reduced pain and number of writhing) (*Chang et al., 2012*).

*In vitro*

The hederacoside C (0.5–5 µg/mL) was shown to decrease the expression of the TLR2 and TLR4 receptors *in vitro* elevated by the *Staphylococcus aureus* infection. Since these receptors recognize the microbial noxious stimuli, their downregulation may modify the extent of the immune response (*Akhtar et al., 2019*). Liquiritigenin and isoliquiritigenin, components of licorice, suppressed eotaxin secretion in the human fetal lung fibroblast. It is therefore possible that the migration of the inflammatory cells may be reduced (*Jayaprakasam et al., 2009*). *trans*-Anethole protected against decreased cell viability, cell barrier injury and increased cell apoptosis, albeit at very high concentrations of 3 and 5 mM. Although these effects were observed in the rat intestinal epithelial cells, the respiratory epithelial cells may respond in a similar manner. Importantly, changes in the expression of 493 genes, most of which were related to antigen processing and presentation, complement and coagulation cascades, and to the NF-κB signaling (see Subsection 5.4.4.), were detected (*Yu et al., 2022*). The stimulation of the macrophage phagocytic activity and migration, and the inhibition of complement activation was reported for the *Althaea officinalis* extract and its polysaccharide fraction (*WHO, 2002*; *Scheffer, Wagner & Proksch, 1991*; *Bonaterra et al., 2020*). The same was described for *Cetraria islandica* products (100 µg/mL) studied in the granulocytes (*Ingolfsdottir et al., 1994*; *Olafsdottir et al., 1999*; *Shrestha, Clair & O'Neill, 2015*). Anti-complement activity was also found in some flavonoids, including kaempferol, kaempferol-3-*O*-rutinoside and quercetin (*Huang et al., 2020*).

*Clinical evidence*

The anti-inflammatory, mucolytic, and bronchodilatory effects were confirmed for 1,8-cineole (200 mg/three doses daily/6 months as a supplementary therapy during winter) in a double-blind, placebo-controlled multi-center-study with 242 patients. Statistically significant improvement of multiple criteria was observed (frequency, duration and severity of exacerbations, lung function, respiratory symptoms, quality of life) (*Worth, Schacher & Dethlefsen, 2009*).

Although the cough is the main objective of this article, the anti-inflammatory effects not directly related to the respiratory system or cough deserve a very short mentioning for the sake of completeness. Positive anti-inflammatory effects of quercetin were found in various clinical trials addressing *e.g.*, rheumatoid arthritis (*Javadi et al., 2017*), polycystic

ovary syndrome (*Vaez et al., 2023*) or hemodialysis patients (*Castilla et al., 2006*). Some effects were also reported for *trans*-anethole (*Hamada et al., 1999*), limonene (*Ostan et al., 2016*; *d'Alessio et al., 2013*), luteolin (*Casetti et al., 2009*; *Lunardelli et al., 2019*), combination of luteolin, quercetin, and rutin (*Taliou et al., 2013*), hydroalcoholic extract of *Glycyrrhiza* spp. root (*Zabihi et al., 2023*) and glycyrrhizic acid (*Cao et al., 2020*). In contrast, other studies reported negligible or no effects on inflammation (*Sajadi Hezaveh et al., 2019*; *McAnulty et al., 2008*; *Heinz et al., 2010*; *McAnulty et al., 2013*; *Hodgin et al., 2021*; *Dehghani et al., 2021*).

## Discussion

The anti-inflammatory effects were reported for both the expectorant and antitussive herbal drugs. The *in vivo* effective doses are usually in the range of units up to low hundreds of mg/kg. Importantly, the studies clearly demonstrated that the anti-inflammatory action can be augmented by other plant components in an additive or synergic fashion. For example, the anti-inflammatory effects of 1,8-cineole were augmented by limonene (a monoterpene from *Eucalyptus*) (*Hirota et al., 2012*; *Chi et al., 2013*). α-Pinene may act in the same way (*Podlogar & Verspohl, 2011*). This synergism is also possible for the combinations of herbal drugs or their components with conventional medicines. For example, *trans*-anethole potentiated the anti-inflammatory effects of ibuprofen in two different rat models of inflammation (paw oedema and pleurisy), and this effect may be mediated at least in part by the TNF-α inhibition (both *trans*-anethole and ibuprofen were given orally in the doses of 62.5 and 8.75 mg/kg, respectively) (*Wisniewski-Rebecca et al., 2015*). The immune-stimulating properties of certain herbal polysaccharides, especially of the β-glucans, should not be disregarded when the anti-inflammatory effects are discussed. While mushrooms and yeasts are their principal sources, some antitussive plants also synthesize these biopolymers. For example, the lichenan from *Lichen islandicus* belongs to the β-glucan family. Unfortunately, no specific data on its effects on immunity or inflammation in the *in vivo* cough models are available. However, certain protective effects of the β-glucans were shown in the rat and murine models of lung injury (oral, i.p. or i.m. administration, doses from 2 to 50 mg/kg) (*Babayigit et al., 2005*; *Bedirli et al., 2007*; *Gulmen et al., 2010*).

Unfortunately, there are virtually no clinical trials studying inflammation of the respiratory system, and most of the available reports deal with other sites of inflammation. The results are inconclusive. Both the reports confirming and not confirming the anti-inflammatory effects can be found. The sites of inflammation studied are very diverse and vary from a reliable model of skin inflammation (UVB irradiation) (*Casetti et al., 2009*) to conditions where the inflammatory etiology is only speculative (*Hodgin et al., 2021*). The methods of evaluation are also very diverse. The determination of the plasma levels of pro-inflammatory markers (TNF-α, IL-6) can be regarded as standard, whereas questionnaire surveys are always, in principle, burdened with a subjective factor. In general, the positive effects are higher if the initial inflammatory status is higher. The herbal anti-inflammatory effects may be beneficial if used as a co-medication with the

standard therapy. In all the reported clinical trials, the investigated anti-inflammatory therapy was very safe.

## Specific mechanisms of the anti-inflammatory effects

The anti-inflammatory effects are mediated by miscellaneous mechanisms, some of which are worth reporting in detail. The text below will describe the principal ones that have been reported for the herbal drugs used to treat the cough or for their active components. Specifically, the effects on the cytokines, effects on cyclooxygenase and lipoxygenase, antioxidant effects, and on the NF-κB signaling will be discussed. For each, a short introduction will again be provided for higher clarity. Individual mechanisms will be commented on in separate subsections. One distinct feature, however, is common to all of them. In the pertinent experiments, the results obtained are strongly dependent on the experimental design and are thus highly inconsistent in some cases, which makes their proper interpretation difficult.

### Effects on the cytokines

Cytokines are the key mediators of the immune response including inflammation (*SinoBiological, 2022*). They constitute a large group of signaling molecules secreted by immune cells. These substances are often produced in a cascade, with one cytokine stimulating the production of another or others. A cytokine is usually secreted by several cell types and also acts on several cell types (pleiotropy). The cytokines influence both the original and the nearby cells (autocrine and paracrine actions) or, under some circumstances, act on distant cells (endocrine action). The same or similar processes can be influenced by various cytokines (redundancy). The effects of the cytokines can be synergic in some cases, but also antagonistic in others. The cytokines are classified according to their origin (lymphokines are produced by the lymphocytes, monokines by the monocytes/ macrophages), their activity (chemokines are chemotactic) and interactions (interleukins (ILs) are secreted by one leukocyte type and act on other leukocytes). There are three large groups of the cytokines: interleukins, interferons and tumor necrosis factors. In general, the cytokines influence cell communications and interactions. They bind at specific receptors on the target cells at picomolar concentrations, and activate intracellular pathways, usually those that belong to protein kinase transduction cascades. The cytokines mediate and regulate immune reactions, including acute or chronic inflammations. Importantly, the cytokines can be classified as pro-inflammatory (*e.g.*, IL-1β, IL-6, tumor necrosis factor-α (TNF-α)) or anti-inflammatory (*e.g.*, IL-1 receptor antagonist, IL-4, IL-10, IL-13, interferon-α (IFNα), transforming growth factor-β). Their orchestrated interplay allows for an effective, but not excessive immune response (*Dinarello, 1997*).

### Experimental methods

The same *in vivo* models of inflammation are used as in the research of inflammation mentioned above (ALI, lung infection *etc.*). The inflammatory models outside the respiratory system are also relatively common. The potent activators of inflammation, mainly IL-1, IL-6 and TNF-α, are the most often studied cytokines. The interpretation of the obtained results is, at least at first glance, somewhat contradictory. While the

suppression of the pro-inflammatory cytokine production is regarded as a protection against an excessive inflammation and tissue damage, their production is, in turn, interpreted as a stimulation of the immune response. For *in vitro* studies, various cell models are used, but not all of them are related to the airways. To induce the inflammatory conditions, LPS is usually applied, while other stimuli, such as histamine, INF-γ or SARS-CoV-2 are scarce (*Li et al., 2018*; *Papies et al., 2021*; *Vigo et al., 2005*).

*Pharmacological studies on the effects on the cytokines*
*In vivo*
The *in vivo* suppression of the pro-inflammatory cytokine production was demonstrated for both the *Hedera helix* extract and its phenolic rich fraction in a murine model of the ALI and in a rat model of arthritis (doses ranging 100–200 mg/kg, in all cases orally administered) (*Shokry et al., 2022a*; *2022b*). Hederacoside C (5–50 mg/kg i.v. or i.p.) had similar effects in two different murine models of inflammation (ALI or inflammation induced by *Staphylococcus aureus* infection) (*Akhtar et al., 2019*; *Han et al., 2022*). Similarly, *trans*-anethole (125–250 mg/kg, i.p. and orally, respectively) protected against the increase in the pro-inflammatory cytokine levels in BALF in the murine model of ALI (TNF-α, IL-1β and IL-6) and in the COPD model (TNF-α and IL-6) (*Kim, Lee & Seol, 2017*; *Kang et al., 2013*). In the former case, the serum cytokine levels were reduced as well. The same was observed in the rat model of pleurisy for oral doses of *trans*-anethole of 250 and 500 mg/kg (*Domiciano et al., 2013*). The licorice extract given orally in the dose of 300 mg/kg in the murine liver inflammation model and menthone administered by inhalation (0.3 up to 1.2 mg/L at 2.8 mL/min for 25 min) in the asthma model displayed similar effects (*Yu et al., 2015*; *Su & Lin, 2022*). Analogously, 1,8-cineole decreased the levels of the principal pro-inflammatory cytokines (TNF-α, IL-1β, IL-6) in three different models of inflammation (ALI, asthma and COPD models). Importantly, the levels of some anti-inflammatory cytokines (IL-10, IL-4, IL-13) also decreased, which is more supportive of an immunomodulatory effect of the monoterpenoid with the concomitant fine tuning of the inflammatory response (*Lee et al., 2016*; *Yu et al., 2018*; *Zhao et al., 2014*).

The *Plantago lanceolata* is probably the most frequently reported antitussive plant in animal studies, investigating the effect on the cytokine production. Its components, plantamajoside and luteolin reduced the pro-inflammatory cytokine levels in the murine and rat inflammatory models (ALI or LPS-induced bronchopneumonia) in the effective doses of 20–100 mg/kg i.p. and orally, respectively. The effects of plantamajoside may be mediated by the depression of the lung TLR4 receptors (*Wu et al., 2016*). On the other hand, the *in vivo* effects of luteolin were accompanied by a reduced miRNA-132 expression in the bronchial tissue. The authors speculated that this small non-coding RNA molecule might act as a mediator for luteolin in its regulation of the LPS-induced inflammation (*Liu & Meng, 2018*). Surprisingly, luteolin reduced the amount of the pro-inflammatory cytokines and exhibited anti-inflammatory effects even upon topical application in rat granuloma and murine ear edema models. These effects were enhanced in the presence of the polysaccharides from *Citrus grandis* Osbeck, suggesting possible potentiation (*Chen et al., 2018*). The reduction of the pro-inflammatory cytokine levels (IL-1β and TNF-α)
was also reported for scopoletin and β-glucan administration in similar doses (1–10 mg/kg i.p., and 50 mg/kg orally, respectively) (*Leema & Tamizhselvi, 2018*; *Chang et al., 2012*; *Babayigit et al., 2005*; *Bedirli et al., 2007*; *Senoglu et al., 2009*). In the case of scopoletin, the modification of the NF-κB signaling was suggested. In contrast to previous studies, an increase in the levels of the pro-inflammatory and anti-inflammatory cytokines after luteolin administration was reported. However, an atypical model of airway inflammation (murine tuberculosis) was used in this study. Regardless of it, the effects of luteolin were considered positive due to the reduced level of necrosis and enhanced long-term protection against tuberculosis mediated by the T and NK cells (*Singh et al., 2021*).

*In vitro*

In accordance with the *in vivo* studies, decreased production of the pro-inflammatory cytokines was found after the incubation with several pure components of the expectorant plants, such as hederacoside C (*Han et al., 2022*), *trans*-anethole (*Chainy et al., 2000*), 1,8-cineole (*Juergens, Stöber & Vetter, 1998*), liquiritigenin (*Kim et al., 2008*), glycyrrhizic acid (*Li et al., 2018*), liquiritin (*Yu et al., 2015*), and three secondary metabolites from *Marrubium vulgare* (*Shaheen et al., 2014*). These results were obtained on diverse cell lines (RAW264.7, J774A.1, THP-1) as well as on the monocytes and lymphocytes from healthy human donors or on the human nasal epithelial cells. The effective concentrations in all cases ranged from 20 to 100 μM. The effects on the cytokine production were dose-dependent and corresponded to the inhibition of mRNA expression (if determined). The dose-dependent decreasing effects on the anti-inflammatory cytokine production were also shown for many antitussive plant components. Among them, the plantamajoside (*Wu et al., 2016*; *Liu et al., 2019*; *Ma & Ma, 2018*; *Son et al., 2017*), luteolin (*Chen et al., 2018*) and scopoletin (*Cheng, Cheng & Chang, 2012*) were the most frequently reported. In all experiments on diverse cell lines, the effective concentrations were roughly in the order of tens of μg/mL. Polysaccharide lichenan increased the secretion of anti-inflammatory IL-10 compared to the IL-12p40, the polymer may thus rather tune the inflammatory response analogously to 1,8-cineole mentioned above (*Freysdottir et al., 2008*). The effects on the cytokine production were also found for various plant extracts such as *Althaea* spp. (*Scheffer, Wagner & Proksch, 1991*), *Pelargonium sidoides* (EPs[®] 7630) (*Papies et al., 2021*; *Kolodziej et al., 2003*; *Witte et al., 2020*), *Cetraria islandica* (*Freysdottir et al., 2008*; *Omarsdottir, Olafsdottir & Freysdottir, 2006*) (effective concentrations roughly 50–500 μg/mL) and for β-glucans (*Murphy et al., 2020*).

*Discussion*

Many *in vivo* and *in vitro* studies reported the possible effects of the cough phytotherapy on the cytokines and their signaling pathways. As mentioned above, the experimental model strongly influences the results. As demonstrated by these examples, the selection of the cell line is crucial. (1) The activated monocytes responded differently to the *Althaea* sp. extracts from the differentiated macrophages or the RAW264.6 cells. Depending on the experimental setup, enhancing, neutral, and decreasing effects on the cytokine production were found (*Kim, Lee & Seol, 2017*; *Scheffer, Wagner & Proksch, 1991*). (2) The epithelial

cells and fibroblasts responded to the polysaccharides from *Althaea officinalis* in very different ways (*Deters et al., 2010*). Although the studies are hardly comparable, (3) the mechanisms modifying the cytokine production are probably very diverse. For example, *trans*-anethole blocks the I-κBα kinase, and thus subsequently decreases the I-κBα (nuclear factor of the kappa light polypeptide gene enhancer in B-cells inhibitor, alpha) degradation (*Chainy et al., 2000*), plantamajoside decreased the expression of the TLR4 receptors (*Wu et al., 2016*), scopoletin was found to down-regulate the cytokine transcription factors GATA-3 and the nuclear factor of the activated T cells (NFAT) with the participation of the protein kinase C (*Cheng, Cheng & Chang, 2012*). The effects of the lichenan may be mediated by DC-SIGN, which is a C-type lectin receptor present on the surface of the dendritic cells and macrophages. For the lichenan, tolerogenic effects similar to those of vitamin $D_3$ were hypothesized (*Freysdottir et al., 2008*). In addition, experiments often focus on just one possible mechanism of action, but a substance can act in multiple ways.

### Effects on cyclooxygenase and lipoxygenase

Since cyclooxygenase (COX) products belong to the principal intermediaries of inflammation, the effects of herbal drugs or their active components on this enzyme are assumed. There are three types of cyclooxygenases: the constitutively expressed COX-1, the inducible COX-2 and the most recently discovered COX-3 which is present in the highest concentrations in the CNS and the heart (*Chandrasekharan et al., 2002*).

In inflamed tissues, COX-2 is highly expressed. COX-2 expression is stimulated by many pro-inflammatory cytokines, growth factors and by endotoxin. Accordingly, COX-2 together with COX-1 expressions as well as the levels of prostanoids were enhanced in the bronchoalveolar lavage of asthmatics (*Lee & Yang, 2013*). 5-Lipoxygenase (5-LOX) is yet another important enzyme which catalyzes the transformation of essential fatty acids into biologically active products, including leukotrienes. The leukotrienes participate in inflammation, allergic and hypersensitive reactions. Among them, the leukotriene $LTB_4$ acts as a chemoattractant for the leukocytes and facilitates their adhesion to the blood vessels in the inflamed tissues. The cysteinyl leukotrienes induce bronchoconstriction, and promote inflammation by increasing vascular permeability and by promoting the activation and migration of leukocytes (*Seibel et al., 2018b*; *Meirer, Steinhilber & Proschak, 2014*).

### Experimental methods

The same inflammation models used in experiments on the cytokines mentioned previously (ALI, paw edema *etc.*), are employed for *in vivo* research. For *in vitro* studies, diverse cell models of inflammation have been developed based on the measurement of cyclooxygenase activity or expression of the related genes. Special biochemical assays are also available. The inducible COX-2 was more frequently investigated than the house-keeping COX-1 isoform. The effects on 5-lipoxygenase were studied using special immune assays.

*Pharmacological studies on the effects on cyclooxygenase and lipoxygenase*
*In vivo*

In the murine model of ALI, administration of the ethanolic *Hedera helix* leaf extract and the phenolic rich fraction (orally 200 and 100 mg/kg, respectively) downregulated the expression of COX-2 (*Shokry et al., 2022a*). Scopoletin, constituent of the Pelargonii radix, decreased induced expressions of COX-2 and iNOS in the murine λ-carrageenan model of the paw edema (*Chang et al., 2012*). A fixed combination of the thyme herb and primula root extracts reduced the levels of $LTB_4$, cysteinyl leukotrienes and $PGE_2$ in BALF in a rat model of the LPS-induced pulmonary inflammation (*Seibel et al., 2018b*).

*In vitro*

The ethanolic extract from *Hedera helix* leaves and its phenolic rich fraction inhibited human recombinant COX-2 activity measured by a commercial colorimetric inhibitor screening assay kit. The latter was slightly more effective (*Shokry et al., 2022a*). Inhibition of both COX isoenzymes was reported for various compounds isolated from *Marrubium vulgare* (*Neamah, Sarhan & Al-Shaye'a, 2018*; *Sahpaz et al., 2002*) and for carvacrol, a monoterpene phenolic constituent of the essential oil from *Thymus vulgaris* (*Landa et al., 2009*). Among antitussives, a hydroalcoholic extract from *Plantago lanceolata* inhibited the COX-2 activity, but not that of lipoxygenase in cell-free systems (*Herold et al., 2003*). Accordingly, plantamajoside decreased the $PGE_2$ levels in the human gingival fibroblasts, stimulated by LPS (*Liu et al., 2019*). On the other hand, no effects of the *Plantago lanceolata* extract on the elevated COX-2 mRNA and the $PGE_2$ levels were detected in the LPS/INF-γ stimulated murine macrophages (*Vigo et al., 2005*). Interestingly, *Althaea rosea* flower extract upregulated the expression of COX-2 and iNOS in the RAW264.6 cells, and these effects were interpreted as immunostimulation (*Kim, Lee & Seol, 2017*).

Inhibition of lipoxygenase was reported for the primula extract, and for various components of the thyme extract; synergistic effects of all compounds are possible. The same was observed for a mixture of dry extracts from the thyme herb and the primula root in the rat whole blood and in the human monocytes and neutrophils stimulated with $Ca^{2+}$-ionophore ($IC_{50}$ roughly units to low tens of μg/mL) (*Seibel et al., 2018b*).

*Discussion*

There are apparent differences in selectivity among the active components in relationship to the inhibition of the cyclooxygenases even from one plant. For example, luteolin-7-*O*-β-glucopyranoside, oleanolic acid and luteolin-7-*O*-rutinoside from *Marrubium vulgare* inhibited preferably COX-1, while apigenin-7-*O*-β-glucopyranoside, luteolin-7-*O*-rutinoside and rosmarinic acid inhibited COX-2 (*Neamah, 2018*). Similarly, COX inhibition was reported for various phenylpropanoid esters from *Marrubium vulgare* and three of them, acteoside, forsythoside B, and arenarioside preferably inhibited the COX-2 (*Sahpaz et al., 2002*).

**Antioxidant effects**

The sensitivity of the sensory C fibers is profoundly increased by the oxidative stress, and the ROS can trigger or intensify the cough. The ROS are associated with the airway

inflammation produced by infiltrating the immune cells as well as within the nociceptive neurons. The mitochondrial production is likely to play a major role. The ROS act as cysteine-modifying reactive electrophiles, and thus modulate a wide range of neuronal signaling cascades *via* ion channels, transporters, enzymes, as well as the signaling molecules themselves and lipids. Among them, certain TRP channels may be of particular importance for the cough (*Nassenstein et al., 2008*; *Taylor-Clark et al., 2008*). Accordingly, the effects of $H_2O_2$ at physiological concentrations are mediated exclusively by the TRPA1 channels. At higher concentrations (exact values depend on the experimental model and species), additional signaling pathways also participate. Notably, the airway TRPA1 channels are also activated by a number of other stimuli, both direct (intracellular $Ca^{2+}$, menthol in low doses, noxious cold, air pollutants) and indirect after the activation of the membrane targets ($G_q$ coupled receptors or, importantly, the TLRs and their downstream pathways) (*Grace et al., 2012*; *Mukhopadhyay et al., 2014*; *Oz et al., 2017*). The TRPV1 channels mediate the respiratory effects of the ROS as well, but higher concentrations are needed (*Taylor-Clark, 2016*). The TRPA1 channel activation in guinea pigs initiates a cough that is relatively modest compared to that initiated by the TRPV1 channel activation. The explanation may be a low efficacy of the TRPA1 channel stimulation in the induction of the sustained activation of the airway C-fibers. This may prevent exaggerated coughing after moderate stimuli (*Brozmanova et al., 2012*). The TRPA1 and TRPV1 channels were reported to be co-expressed (*Taylor-Clark et al., 2008*). The channels thus might interact and produce synergic response (*Spahn, Stein & Zöllner, 2014*).

*Experimental methods*

Elevations of the serum superoxide dismutase (SOD), catalase (CAT), and glutathione peroxidase (GPx), or decreases of the malondialdehyde (MDA) and myeloperoxidase (MPO) levels are most frequently measured variables for the antioxidant effect quantification. The MPO serves not only as a marker of oxidative stress, but also as that of neutrophil accumulation in the inflammatory tissue. In *in vitro* studies, various additional methods for the evaluation of the antioxidant properties were applied, including the measurement of the total-oxidant capacity, scavenging effects, ferric-reducing antioxidant power, modulation of the respiratory burst, and others. The results should be treated with caution, as they may easily lead to misleading conclusions. For example, an increase in the levels of antioxidant enzymes may be an adaptor response to the oxidative stress, and ferric reduction can be associated with pro-oxidation (*Dodson, Castro-Portuguez & Zhang, 2019*; *Chen et al., 2015*; *Macáková et al., 2012*).

*Pharmacological studies on the antioxidant effects*

*In vivo*

Antioxidant effects or prevention of oxidative stress were found for hederacoside C and 1,8-cineole in the doses of roughly units to tens of mg/mL, i.p. and orally, respectively (*Akhtar et al., 2019*; *Zhao et al., 2014*), and for the *Hedera helix* extracts administered orally in the doses of 100 and 200 mg/kg (*Shokry et al., 2022a*). Among antitussive plants, the antioxidative effects were found for scopoletin (*Leema & Tamizhselvi, 2018*), for a

secondary metabolite from *Cetraria islandica* (*de Barros Alves et al., 2014*), for β-glucans (*Bedirli et al., 2007*; *Gulmen et al., 2010*), and for the extract from *Pelargonium sidoides* (EPs® 7630) (*Bao et al., 2015*).

*In vitro*

In *in vitro* experiments, the antioxidant effects were observed mainly for licorice and its constituents. This antioxidant activity is attributed to the phenolic content of licorice, with flavonoids and isoflavones being the main candidates as the active substances (*Yu et al., 2015*; *Kim et al., 2008*; *Racková et al., 2007*; *Sharma & Pandey, 2013*; *Singh, Pal & Darokar, 2015*). An antioxidant activity was also reported for the extract from *Primula veris*. Notably, higher antioxidant effects were found in the flowers rather than the roots (both in fresh and dried extracts), and their intensity was directly proportional to the total phenolic content (*Tarapatskyy et al., 2021*). Among antitussives, *Plantago lanceolata* and its principal constituent, plantamajoside, probably have certain antioxidant properties as well (*Herold et al., 2003*; *Choi et al., 2008*). The same is true for the *Althaea officinalis* root extract and fraxetin (a compound of Pelargonii radix) (*Bonaterra et al., 2020*; *Chang et al., 1996*).

*Discussion*

Some protective antioxidant effects of the cough herbal drugs or their components were experimentally shown. In animal models, the effective doses were roughly in the order of tens of mg/kg given i.p. or orally. In one study, intramuscular administration of β-glucan in two doses of 2 mg/kg decreased the lung MPO activity in the rat inflammation model (*Bedirli et al., 2007*). The mechanism of the antioxidative action is not very clear, and probably involves various intracellular targets. In the case of the *Hedera helix* extracts, the increased expressions of p38-MAPK (mitogen-activated protein kinase) and phospho-p38-MAPK were attenuated, which implies the participation of this signaling pathway (*Shokry et al., 2022a*). Glycyrrhizic acid, liquiritin and liquiritigenin inhibited the LPS-stimulated NO production in a dose-dependent manner (*Yu et al., 2015*). Accordingly, liquiritigenin was found to prevent the iNOS induction, albeit at very high concentrations (3–30 mM) (*Kim et al., 2008*). Three secondary metabolites of *Marrubium vulgare* had similar effects (*Shaheen et al., 2014*). The antioxidant activity of *Plantago lanceolata* is also hypothesized to be mediated by NO-scavenging or by the inhibition of the iNOS gene expression (*Vigo et al., 2005*). The effects of glabridin, isoflavan from the licorice root were mediated by the elevated expressions of various genes involved in the response to the oxidative stress. Notably, glabridin had antioxidant effects at low concentrations, while at higher concentrations, this isoflavan induced the generation of intracellular ROS and NO (*Singh, Pal & Darokar, 2015*). Hence, the antioxidant effects are manifested only within a certain dose or concentration ranges, and can be reversed to prooxidant effects at higher concentrations.

Unfortunately, the data on the plant polysaccharides and their antioxidant effects in the cough treatment are scarce. The effects of the orally administered β-glucans outside the airways were reported (*Senoglu et al., 2009*). However, it is well known that the polysaccharide composition, conformation and molecular weight influence the

antioxidant activity. Antioxidant properties become more pronounced with increased content of phenolic acids and proteins (polysaccharide-polyphenol and polysaccharide-protein covalent conjugates and non-covalent complexes), with selenylation and derivatization, such as sulfation, carboxylation, phosphorylation, benzoylation, acetylation or oxidation. Chemical modification is often accompanied by a decrease in molecular weight which is also regarded as positive for the antioxidant potential (*Nemzer et al., 2019*).

### Effects on the NF-κB signaling

Among the pro-inflammatory signaling pathways, the transcription factor nuclear factor κB (NF-κB) signaling is one of the most important and the best understood. The NF-κB regulates a large group of genes involved in the inflammatory response. In mammals, it is composed of five factors (p50, p52, p65, c-Rel and Rel-B). Under normal circumstances, inhibitory proteins, mainly the I-κB family, regulate its activity. Activation of NF-κB is triggered by two principal pathways. The canonical (classical) pathway is activated upon activation of various receptors, including pattern-recognition receptors, receptors for certain cytokines, TNF receptors as well as the T- and B-cell receptors. This activation leads to the degradation of I-κB mediated by the I-κB kinase. Rapid translocation of NF-κB into the cell nucleus and regulation of the gene transcription influence almost all components of the immune response. The non-canonical (alternative) NF-κB pathway is supplementary, and does not involve the I-κB degradation (*Liu et al., 2017*).

NF-κB mediates proliferation, differentiation and regulation of innate and adaptive immune cells, such as macrophages, dendritic cells and neutrophils, and inflammasomes (*Hayden & Ghosh, 2011*). The innate immune cells express various pattern-recognition receptors, including the TLRs through which they recognize the PAMPs and DAMPs, and selectively activate the canonical NF-κB pathway (*Kawai & Akira, 2010*; *Kumar, Kawai & Akira, 2011*; *Takeuchi & Akira, 2010*). Activation of the NF-κB responsive genes leads to early production of antimicrobial substances, which include the ROS and NO, the latter through iNOS activation (*Farlik et al., 2010*). Notably, the ROS can both activate and repress the NF-κB signaling in a phase- and context-dependent manner (*Lingappan, 2018*). The pro-inflammatory mediators, such as IL-1, IL-6, IL-12 and TNF-α are other important products of the NF-κB signaling. These mediators promote inflammation directly by the recruitment and activation of the immune cells, and indirectly by supporting the inflammatory T-cells differentiation. The non-canonical NF-κB pathway is involved in the development of the secondary lymphoid tissues (lymph nodes, Peyer patches, spleen), and is necessary for the optimal function of the adaptive immune system. Last, but not least, NF-κB regulates inflammasomes. These intracellular multi-protein complexes assemble in response to the PAMPs and DAMPs, and contribute to the activation of the inflammatory caspases.

The inflammation is beneficial only if its extent corresponds to the noxious stimulus. NF-κB participates both in the inflammation onset and in its physiological down-regulation. Thus, NF-κB mediates macrophage differentiation into the classically activated M1 macrophages which are pro-inflammatory, and into activated M2 macrophages which

produce anti-inflammatory cytokines, such as IL-10 and IL-13. Similarly, the canonical NF-κB regulates CD4+ T-cell differentiation; out of them, the Th1 and Th17 are the principal inflammatory T cells. The regulatory $T_{reg}$ cells are also produced, controlling the adequate extent of the inflammation. Differentiation into a certain T-cell subtype probably depends on the cytokine environment (*Hayden & Ghosh, 2011*). Accordingly, inhibition of the NF-κB can protract the inflammatory response and prevent proper tissue repair (*Lawrence et al., 2001*). Additionally, a dysregulation of the NF-κB activation is suspected to promote some autoimmune and inflammatory diseases (*Pai & Thomas, 2008*).

*Experimental methods*

A possible suppression of the NF-κB signaling by cough phytotherapy was investigated in the same animal inflammation models as those described above. The *in vitro* experiments on cell cultures were more frequent. From the parameters reported, the amount of the NF-κB and the I-κBα proteins, translocation of the NF-κB into the nucleus, expressions of the related genes and the related protein levels are the most common.

*Pharmacological studies on the effects on the NF-κB signaling*

*In vivo*

In two different murine models of airways inflammation (ALI or *Staphylococcus aureus* infection), the stimulation of the NF-κB signaling was diminished after the administration of hederacoside C (doses of roughly tens of mg/kg, i.v. and i.p., respectively). Importantly, the up-regulated levels of phosphatidylinositol 4,5-bisphosphate ($PIP_2$), diacylglycerol and inositol-1,4,5-trisphosphate ($IP_3$) and down-regulated level of PLCγ2 were subsequently also modified, indicating a participation of this signaling pathway (*Akhtar et al., 2019*; *Han et al., 2022*). Similar effects were observed for 1,8-cineole in oral doses of 10–100 mg/kg in the murine model of ALI (*Zhao et al., 2014*). From antitussives, a decrease in the NF-κB signaling was produced by i.p. plantamajoside, luteolin and scopoletin in the inflamed lungs (*Wu et al., 2016*), and in the model of the cerulein-induced acute pancreatitis linked to a lung injury. Effective doses were roughly in the order of tens of mg/kg (*Liu & Meng, 2018*).

*In vitro*

Suppression of the stimulated NF-κB signaling *in vitro* was observed for the expectorant hederacoside C (*Akhtar et al., 2019*; *Han et al., 2022*), *trans*-anethol (*Yu et al., 2022*; *Aggarwal & Shishodia, 2004*; *Sen, Traber & Packer, 1996*), 1,8-cineole (*Greiner et al., 2013*), glycyrrhizin (*Li et al., 2018*) and liquiritigenin (*Kim et al., 2008*) in diverse cell cultures. The same was reported for active constituents of *Plantago* spp.: plantamajoside (*Wu et al., 2016*; *Liu et al., 2019*; *Ma & Ma, 2018*; *Son et al., 2017*; *Han, Nam & Lee, 2016*), luteolin (*Liu & Meng, 2018*; *Chen et al., 2018*) and apigenin (*Zhang et al., 2014*), and for the coumarins from Pelargonii radix: scopoletin and fraxetin (*Hassanein et al., 2020*). In all these experiments on the cell cultures, the effective concentrations were roughly in the order of units of μg/mL or tens of μM. The effects of *trans*-anethole may be mediated by the TLR4 receptor (reported in rat intestinal epithelial cells) (*Yu et al., 2022*).

*Discussion*

*In vivo* suppression of the NF-κB signaling is produced in the doses roughly of tens of mg/kg, regardless of the way of administration. While these effects are evident, their mechanism remains less clear. Modulations of the NF-κB/p65 and I-κBα levels and localizations are probably crucial. For example, glycyrrhizin inhibited histamine-induced mRNA expression and secretion of NF-κB/p65 and I-κBα in the human nasal epithelial cells (*Li et al., 2018*). Similarly, hederacoside C suppressed NF-κB/p65 and I-κB-α expressions in the RAW264.7 cells exposed to *Staphylococcus aureus* (*Akhtar et al., 2019*), while the substance prevented the translocation of NF-κB/p65 into the nucleus in the LPS-stimulated macrophages (*Han et al., 2022*). A total of 1,8-cineole exhibited similar effects in the human peripheral blood mononuclear cells, in the human glioblastoma U373 cells and the cervical cancer HeLa cells (*Greiner et al., 2013*). As reported for hederacoside C and liquiritigenin, both the expressions of NF-κB p65 and I-κB and the phosphorylation of these proteins can be inhibited (*Akhtar et al., 2019*; *Kim et al., 2008*). Plantamajoside-induced decrease in the expressions of I-κBα and of NF-κB/p65 may be interlinked with the mitogen-activated protein kinase (MAPK) signaling pathway (*Wu et al., 2016*; *Liu & Meng, 2018*). Notably, luteolin was simultaneously reported to decrease the miRNA-132 expression, which may constitute a perspective topic for further pharmacological research (*Liu & Meng, 2018*; *Dong et al., 2021*).

## Anti-infective effects

An infection of the upper or lower respiratory tracts often results in the cough-provoking inflammation and represents probably the most common cause of cough at all. Both antibacterial and antiviral effects of the cough phytotherapy are thus investigated. More detailed information on the course of the bacterial and viral airway infections is beyond the scope of this review, and the reader can be referred to available review papers or books (*e.g.*, (*Rai et al., 2021*; *Peterson, 1996*)).

*Experimental methods*

*In vivo* research was performed on various animal models of infection. The evaluation was very diverse—from simple monitoring of the animal health status to the histological analysis of the lungs. The biochemical parameters associated with inflammation were also often measured (some of them have already been mentioned above, for example the levels and expressions of the cytokines and other pro-inflammatory proteins). For the research on antibacterial and antiviral activity, the *in vitro* cell models were used more frequently, and, hence, bacterial growth inhibition and MIC values were usually determined. Evaluation of the pathogen adhesion constituted yet another possibility. Interestingly, the changes were always interpreted as protective—a decrease in the adhesion was considered as the prevention of the infection, while the opposite as the facilitated elimination of the pathogen. The measurement of the phagocytic capacity of the inflammatory cells in the culture medium in the presence of a specific bacterial strain was also used. Unfortunately, the biochemical mechanisms of the antibacterial effects or specific targets were rarely reported. Of interest can be the results on the synergism of several natural substances or

when combined with standard clinically used antibiotics. The research on antiviral effects focuses mostly on the influenza virus and rhinovirus. Various methods were used, including antiviral activity assays, neuraminidase and hemagglutination inhibition assays or immunofluorescence monitoring of specific antibodies.

### Pharmacological studies on the antibacterial effects

The data on the *in vivo* antibacterial activity of the cough phytotherapy are scarce. Of potential interest is the activity of luteolin, a *Plantago* constituent, against *Mycobacterium tuberculosis* (5 mg/kg i.p.) with the potentiation of the effects of isoniazide therapy and reduction in the treatment length (*Singh et al., 2021*).

In *in vitro* experiments, some antimicrobial activity was found for the *Hedera helix* phenolic rich fraction; the MIC values against *Klebsiella pneumonia, Mycoplasma pneumoniae* and *Streptococcus pneumoniae* were in units of µg/mL. Importantly, these values were comparable to those of ciprofloxacin in the same experiment (*Shokry et al., 2022a*). Similarly, licorice and its constituents, glabridin and liquiritigenin, showed antibacterial effects at the concentrations of one to low tens of µg/mL. Liquiritigenin and isoliquiritigenin inhibited the growth of *Mycobacterium tuberculosis*, and isoliquiritigenin also that of *Mycobacterium bovis* (MIC at 25 and 50 µg/mL, respectively) (*Singh et al., 2021*; *Singh, Pal & Darokar, 2015*; *Brown et al., 2007*; *Chokchaisiri et al., 2009*). On the other hand, the effects of isoliquiritigenin against *Staphylococcus aureus, Staphylococcus epidermidis*, and *Staphylococcus hemolyticus* were relatively weak ($MIC_{50}$ at 125 µg/mL) (*de Barros Machado et al., 2005*). The same was true for the antibacterial effects of various *Primula* spp. extracts. However, their activity against *Pseudomonas aeruginosa* with the MIC value roughly 0.5–1 mg/mL depending on the species and extract used, is worthy to mention (*Khan et al., 2022*).

Importantly, a synergism in antibacterial effects of glabridin with some clinically used antibiotics, including norfloxacin, oxacillin and vancomycin, was observed (*Singh et al., 2021*; *Singh, Pal & Darokar, 2015*). Additive or weak synergism in the antibacterial effects against *Streptococcus pneumoniae* was also found for the combinations of the anise essential oil with amoxicillin or ciprofloxacin. In this essential oil, *trans*-anethole was identified as its major constituent (90%) (*Gradinaru et al., 2014*). Unfortunately, the activity of the *Foeniculum vulgare* fruit essential oil (*trans*-anethole content 81%) may be accompanied by relatively high toxicity ($LC_{50}$ 10 µg/mL in the Brine Shrimp Lethality Test) (*Akhbari et al., 2019*). Synergism was also hypothesized for *Eucalyptus* constituents including 1,8-cineole and aromadendrene (*Mulyaningsih et al., 2011*). On the other hand, although there is evidence of some antibacterial activity for the fractions obtained from the extracts of *Marrubium vulgare*, the effective concentrations are high (mg/mL range) (*Djerrou, 2015*). Among antitussive herbal drugs, the extracts from *Pelargonium sidoides* (EPs® 7630) and from *Althaea officinalis* exhibited weak antibacterial activities as well (*Rezaei et al., 2015*; *Kolodziej et al., 2003*).

## Discussion

It is necessary to point out that the experimental design or effective concentrations in some *in vitro* studies raise certain doubts, in particular because of the selection of the bacterial strains investigated or the high concentrations needed for the inhibition. Some studies indicate that an extract or pure substances were obtained first, and then randomly tested on the models that were easily available. The clinical relevance of the data provided by these experiments is unclear. Moreover, the antibacterial activities were, in some cases, accompanied by a relatively high toxicity (*Akhbari et al., 2019*), which apparently hinders the potential for, at least, systemic use. In some *in vitro* studies, this important information is entirely omitted. In short, although the research on the antibacterial activity is easy to justify, it is questionable (1) whether the required concentrations are achievable and sustainable with the phytotherapy, and, if so, (2) whether they are safe.

The mechanisms of the antibacterial action in the cough phytotherapy remain often unelucidated, and are probably very diverse. For example, the antibacterial effects of glabridin against *Staphylococcus aureus* are attributed to the generation of oxidative stress, DNA fragmentation and protein degradation (*Singh et al., 2021*; *Singh, Pal & Darokar, 2015*). Acteoside, present in *Plantago lanceolata*, enhanced the INF-γ production at both translational and transcriptional levels (*Song et al., 2016*). The presence of *trans*-anethole in the culture medium increased the phagocytosis of *Staphylococcus aureus* by the neutrophils. This effect may be mediated by the increased production of the IL-8 (important chemoattractant) (*Kwiatkowski et al., 2020*). The extract from *Pelargonium sidoides* root (EPs® 7630) modified bacterial adhesion and improved the epithelial cell differentiation (*Conrad et al., 2007*; *Roth, Sun & Tamm, 2021*). The bacteriostatic effects of some coumarins were attributed to their interactions with the cell membrane (*Kayser & Kolodziej, 1999*). Flavonoids, and in particular their structural congeners chalcones, decreased the expression of some bacterial genes, inhibited bacterial growth and reduced the production of bacterial toxins (*Wang et al., 2015*).

### Pharmacological studies on the antiviral effects

The activity against the influenza virus was found for i.p. acteoside, a constituent of *Plantago lanceolata*, in the murine model. The effective dose was 80 mg/kg (*Song et al., 2016*). The same was suggested for the inhaled *Pelargonium sidoides* root extract (EPs® 7630) (*Theisen & Muller, 2012*). Interesting data showed effective anti-influenza *in vitro* activity for *Glycyrrhiza* spp. and for *Pelargonium sidoides*. *Glycyrrhiza* spp. contains more than 20 triterpenoids and nearly 30 flavonoids. Out of the triterpenoids, glycyrrhizin and glycyrrhetinic acid have been reported to possess antiviral effects. The mechanism of their antiviral action is probably complex. The compounds may inhibit viral gene expression and replication, reduce the adhesion force and stress, and reduce the binding of HMGB1 (high-mobility group box one protein which supports transcription of many genes) to DNA. They may also block the degradation of I-κB, activate T lymphocyte proliferation and suppress the host cell apoptosis. The antiviral activity of the *Pelargonium sidoides* extract (EPs® 7630) may include the following three effects: (1) up-regulation of the vitamin D receptor expression and its translocation into the nucleus. In this way, the

anti-viral effects of calcitriol against Rhinovirus-16 were enhanced; (2) improved differentiation of the human bronchial epithelial cells (*Roth, Sun & Tamm, 2021*); (3) impairment of the viral hemagglutination and neuraminidase activities – prodelphinidins could be responsible for this effect (*Theisen & Muller, 2012*). Accordingly, the combination of the *Pelargonium sidoides* root extract (EPs® 7630) and the antiviral drug zanamivir produced the synergistic inhibition of neuraminidase (however, these results were obtained on the bacterial *Vibrio cholerae* neuraminidase) (*Quosdorf, Schuetz & Kolodziej, 2017*).

## Other mechanisms possibly participating in the cough phytotherapy

In the text below, the data on the indirect, weak or not well documented effects of the cough phytotherapy are provided. These effects are probably less important than the above-mentioned, but should not be omitted.

### Effect on $\beta_2$ adrenoreceptors

The $\beta_2$ adrenoreceptors are a well-known target of bronchodilator sympathomimetics, such as salbutamol (albuterol). Thus the question whether the herbal therapy could induce similar effects can be raised. α-Hederin (1 μM, 24 h) caused the activation of the $\beta_2$-receptors on the human embryonic kidney HEK293 cells with an increase in the intracellular cAMP levels. This activation could be linked to both the surfactant production by the alveolar type II cells and the smooth muscle relaxation (*Sieben et al., 2009*). α-Hederin increased the isoprenaline-induced relaxation of the bovine tracheal smooth muscle strips pre-contracted with metacholine. Inhibition of the heterologous desensitization of $\beta_2$-adrenoreceptors was also considered (*Wolf et al., 2011*). On the other hand, no $\beta_2$-sympatomimetic activity was found for hederacoside C and hederagenin. Spasmolytic activity of the *Hedera helix* dry extract was confirmed *in vitro* on the isolated guinea pig ileum. Given the high saponin content, the authors regarded saponins as the components most responsible for these effects, but no conclusive evidence was provided (*Trute et al., 1997*).

### Effects on ion channels

As already mentioned, the airway TRP channels sense a huge variety of stimuli, including chemicals, changes of osmolarity and pH, or mechanical stretch in a stimulus-specific fashion. For example, in the lower respiratory tract, the TRPV1 channels are involved in the sensing of heat, while the TRPM8 channels in the sensing of cold. There is a possibility of a direct modulation of these TRP channels, which would result in the peripheral anesthetic-like action on the sensory nerves and less frequent triggering of the cough reflex.

Little information is available to date, with the exception of some data on 1,8-cineole. This terpenoid ether activated the cool temperature-sensitive TRPM8 channels and inhibited the cold-sensing TRPA1 channels. These effects may have led to the analgetic effect (*Takaishi et al., 2012*). Importantly, the effects of 1,8-cineole on the tracheal smooth muscle *in vitro* are dose-dependent and bell-shaped. Low doses of 1,8-cineole potentiated contractions, and this effect was likely mediated by the activation of the TRPM8 channels. On the other hand, high doses blocked the voltage-gated L-type calcium channels and led

to the muscle relaxation (*Pereira-Gonçalves et al., 2018*). This dose-effect relationship should be taken into consideration if further investigation is carried out.

### Effects on adhesion molecules

Adhesion molecules regulate cellular functions, tissue integrity and homeostasis. They mediate the interactions among cells, modulate intracellular signaling and also provide structural support for the extracellular matrix. The intercellular adhesion molecule 1 (ICAM-1) and the vascular cell adhesion molecule 1 (VCAM-1) are the most frequently mentioned ones. ICAM-1 is a cell surface glycoprotein that is expressed at low levels in the immune, epithelial, and endothelial cells, and up-regulated in response to inflammation. Its expression is induced by many pro-inflammatory cytokines, such as IL-1β and TNF-α through activation of NF-κB. The glycoprotein regulates leukocyte trafficking and trans-endothelial migration, and modifies immune cell functions. It is highly expressed in the inflammatory macrophages, and mediates the binding of apoptotic cells. In addition, the biomolecule also modulates intracellular signaling (*Bui, Wiesolek & Sumagin, 2020*). In the airways, the pro-inflammatory cytokines cause ICAM-1 up-regulation, and thus elevate the neutrophil-epithelial cell adhesion (*Tosi et al., 1992*). Accordingly, ICAM-1 expression was increased due to bronchial secretion in the humans with bronchiectasis, and this increase correlated with the levels of TNF-α (*Chan et al., 2008*). An increase in the airways epithelial permeability and augmented neutrophil migration mediated by ICAM-1 were also confirmed in the human bronchial epithelium (Calu-3 cells), and in the cow tracheal epithelial cells *in vitro* (*Choi, Fleming & Serikov, 2007*). ICAM-1 is also up-regulated in the patients with a chronic airflow obstruction (including COPD) (*Shukla, Hansbro & Walters, 2017*). Similarly, VCAM-1 is an inducible cell transmembrane glycoprotein, which was identified on the endothelial cells, but is expressed on several cell types. Its expression is induced under inflammatory conditions, including the bronchial asthma and infection. The molecule enhances the transmigration of the polymorphonuclear leukocytes. In asthmatic patients, the entry of the eosinophils into the lungs is VCAM-dependent (*Chin et al., 1997*; *Lee et al., 2013*).

Very few studies on the effect of the cough medicinal plants on the adhesion molecules were reported. In a rat model of COPD, a long-term oral administration of 1,8-cineole in the dose of 260 mg/kg resulted in alleviated ICAM-1 expression in BALF (*Yu et al., 2018*). In the cell cultures, plantamajoside (10 μM) significantly suppressed adhesion molecule expression and monocyte adhesion to the human umbilical vein endothelial cells treated with glycer-AGEs (*Son et al., 2017*). The *Pelargonium sidoides* root extract (EPs® 7630) diminished the adherence (to ca 45%) of a group *A Streptococcus* (GAS) to the human laryngeal epithelial cells at the effective concentration of 30 μg/mL. The authors suggested interactions with the binding factors on the bacterial surface, in which proanthocyanidins may play the main role. An interaction of prodelphinidins with the streptococcal binding sites in a specific manner was proposed (*Janecki et al., 2011*). The prodelphinidins from the *Pelargonium sidoides* root extract (Eps® 7630; 10–100 μg/mL) were also reported to inhibit SARS-CoV-2 entry into the human bronchial epithelial cells Calu-3, with both the endosomal and the plasma membrane fusion-mediated entry processes having been

affected (*Papies et al., 2021*). In the human bronchial epithelial cells infected with rhinovirus, the *Pelargonium sidoides* root extract (EPs® 7630) down-regulated the cell membrane docking proteins and up-regulated the host defense proteins β-defensin-1 and SOCS-1. Surprisingly, no changes in ICAM-1 expression were induced in the same experiment (*Roth et al., 2019*).

### Immunomodulatory effects on the extracellular matrix

The thin protective layer of the herbal mucilage covering the surface of the upper respiratory tract does not provide just a simple mechanical protection, but, surprisingly, may also influence the extracellular matrix, principally the hyaluronic acid. Hyaluronic acid is a negatively charged non-sulfated linear polymer, composed of monosaccharides, D-glucuronic acid and $N$-acetyl-D-glucosamine, linked *via* alternating β-(1→4) and β-(1→3) glycosidic bonds. Its molecular weight can be as high as $10^7$ Da. This macromolecule is physiologically produced by hyaluronan synthases. Every day, roughly one-third of the hyaluronic acid content is degraded and newly synthesized. The degradation of the polymer into low molecular weight fragments is secured by the action of hyaluronidases. Six human hyaluronidases have been described, among them the Hyal-1, Hyal-2 and PH-20 being the most important. Traditionally, hyaluronic acid is regarded as the main component of the extracellular matrix, which contributes to tissue hydrodynamics through binding water and regulating the osmotic pressure. It is currently evident that hyaluronic acid plays an important role in regulation of the innate immunity system (*Lennon & Singleton, 2011*). The compound influences the migration and proliferation of cells, such as the leukocytes and macrophages, and modifies the interactions of the cell surface receptors. The effects of hyaluronic acid on the immune system are principally mediated by binding to the TLRs (mainly TLR2 and TLR4), often (but not necessarily) in the presence of the endogenous receptor for hyaluronic acid CD44. Additionally, the number of the immune receptors can be augmented by the induction of their expression or by their mobilization on the cell membrane. Importantly, the immunological effects of hyaluronic acid are strongly influenced by its molecular weight. The low molecular weight molecules (<500 kDa) stimulate the TLR-mediated inflammation, invasiveness and angiogenesis, enhance the endothelial cell migration, and reduce the water-binding capacity (*Sendker et al., 2017*). In the macrophages, they induce the expression of important pro-inflammatory cytokines, IL-1β and TNFα, and activate the NF-κB/I-κBα autoregulatory pathway. The presence of the high molecular weight hyaluronic acid (>500 kDa) impedes these effects and promotes the epithelial resistance to injury through the NF-κB activation mediated again by the TLRs (*Tighe & Garantziotis, 2019*), resulting in anti-inflammatory, immuno-suppressive, anti-angiogenic and space filling effects. The final outcome is determined by the balance between the low and high molecular weight hyaluronic acids. This balance, in turn, reflects that between the synthesis and degradation of the hyaluronic acid. The degradation rises up in response to a local tissue injury or stress. Hypoxia increases both the synthesis and degradation, which may promote angiogenesis as a compensatory mechanism (*Gao et al., 2005*). The hypoxia also upregulates the expression of the CD44 receptors (*Krishnamachary et al., 2012*). Increased

levels of the low molecular weight hyaluronic acid are present in the lungs in virtually all respiratory diseases (*Lauer et al., 2015*). Deceleration of hyaluronic acid degradation may arise as an interesting new possibility for the fine tuning of the immune response in the respiratory tract.

*Experimental methods*
Suitable methods are yet to be developed. To date, special assays and gene expression measurements have been reported in research articles.

*Pharmacological studies on the immunomodulatory effects on the extracellular matrix*
Very little is known about the effects of the plants used in the cough therapy on the extracellular matrix of the upper airways. The extracts from Althaeae radix and Malvae sylvestris flos showed certain *in vitro* inhibition of the Hyal-1. $IC_{50}$ was in the order of units of mg/mL (*Sendker et al., 2017*; *Orlando et al., 2015*). No data from *in vivo* animal models are available, and hence the real significance of this effect is somewhat speculative.

### Effects on cortisone
Hydrocortisone is converted to cortisone in the human lung tissue by the action of 11β-hydroxysteroid dehydrogenase 2. The reaction is strongly inhibited by glycyrrhetinic acid from *Glycyrrhiza* spp. with the $IC_{50}$ of approx. 25 nM and may, probably just theoretically, partially explain its anti-inflammatory action (*Schleimer, 1991*). On the other hand, it should be noted that the inhibition of this enzyme by glycyrrhetinic acid is associated with the development of a reversible hypertension (*Mladěnka et al., 2018*).

## CONCLUSIONS
Cough represents a common symptom of many acute and chronic diseases. Medicinal plants have been used for the cough therapy widely around the world and their use is based primarily on traditional experience. Confirmation of these effects by scientific studies is thus desirable, but, unfortunately, has not yet been sufficiently achieved in many cases. From the data available, the following conclusions can be drawn:

1) The profit of phytotherapy in the treatment of cough is beyond doubts. In some plants, expectorant or antitussive effects at clinically attainable doses have been reliably documented. *In vivo* animal models have an indispensable place in this research. *In vitro* experiments play an important role in the detection of the possible mechanisms of action, but cannot be considered as definitive evidence of effectiveness under real clinical conditions.

2) In phytotherapy, more simultaneous effects are usually displayed. In addition to the two main effects, expectorant or antitussive, others can also be manifested. This fact can be considered as one of the advantages of the phytotherapy. The constituents of herbal drugs can thus not only relieve the cough, but also target its cause, which is mostly the inflammation of the respiratory tract due to an infection. The phytotherapy is thus based on the interplay or synergism of diverse pharmacological activities. Among them, the anti-inflammatory activity is probably the most important one.

3) On the other hand, this complexity makes the identification of the main active substance difficult. Moreover, the contents of the herbal drugs can vary depending on a number of factors, which can complicate the analysis.

4) If cough arises as a secondary symptom of a chronic disease such as asthma or COPD, correct diagnosis and conventional therapy as soon as possible are necessary. The same holds true in the case of acute bacterial infections, for which appropriate antibiotics are the cornerstone of the treatment. However, boosting of the conventional treatment with phytotherapy in all patients may be advantageous. In chronic diseases, the phytotherapy can reduce the maintenance dose of the conventional therapy, reduce the severity of the symptoms, and improve the quality of life. In acute diseases, the phytotherapy can alleviate the course and shorten the duration of the disease.

5) When phytotherapy and conventional therapy are combined, a synergism of positive effects can occur. This possibility is of great interest for further research and even for the development of combination medicines, for example with antibiotics or NSAIDs.

6) A variety of interesting mechanisms is described, including intracellular effects on the protein synthesis, particularly in the context of suppression of the inflammation. However, it should be noted that the changes in the gene expression and protein transcription are somewhat delayed, and it is questionable whether and to what extent they would be effective in the short-term treatment of common colds. In this respect, medicinal plants should be considered rather as an inspiration for the development of new anti-inflammatory agents with applications in long-term indications.

7) Some of the proposed effects of phytotherapy, such as effects on the extracellular matrix or adhesion molecules, are poorly documented, and deserve further attention.

The way scientific research in this field itself is conducted is also worth commenting. At first glance, a plethora of pharmacological studies on the cough phytotherapy, both expectorants and antitussives, is available. However, their results should be interpreted with caution. Not all results are of sufficient relevance to justify further animal testing or even clinical use. It is obvious that the choice of the pharmacological model depends on the aim of the study and on the issue to be dealt with. Even so, some experiments are unnecessarily methodologically inconsistent, which makes them difficult to compare, let alone interpret the results. In addition, the research is not always carried out systematically. For example, using one or two cell lines to evaluate the effects of a substance, and then drawing a definite conclusion, is insufficient. Moreover, toxicity is yet another factor that has not been investigated in some cases. The reports focused on the underlying biochemical mechanisms suffer from the same drawbacks. Finally, some studies examined unrealistic doses or concentrations or, more than that are pharmacologically questionable to say the least. For example, direct effects on the β-adrenergic receptors were excluded in the case of the *Althaea officinalis* extract (*Alani, Zare & Noureddini, 2015*), but the study was performed on the rat trachea which is not the appropriate model for the β-adrenergic receptor experiments (*Rajani, Shah & Gulati, 1977*).

In conclusion, the conventional cough therapy is certainly not fully substitutable by the phytotherapy, but the latter represents a useful alternative. Accessibility without having to consult a physician is another advantage. In the future, more detailed studies on the effect of the natural extracts or isolated constituents and the elucidation of the molecular mechanism(s) as well as possible toxicity are desirable in order to promote the use of cough phytotherapy alone or in combination with the conventional pharmacological remedies. The use of appropriate scientific methodologies with adequate standards is strongly advocated. Last but not least, the medicinal plants represent a very inspirating source for the discovery of new lead molecules for future drug development.

## ACKNOWLEDGEMENTS

Dr. Russell Kitson is gratefully acknowledged for language corrections.

### Funding

This open-access review article was supported by the Erasmus+ Programme of the European Union, Key Action 2: Strategic Partnerships, Project no. 2020-1-CZ01-KA203-078218 and Charles University (SVV 260 549). The funders had no role in study design, data collection and analysis, decision to publish, or preparation of the manuscript.

### Grant Disclosures

The following grant information was disclosed by the authors:
European Union, Key Action 2: Strategic Partnerships: 2020-1-CZ01-KA203-078218.
Charles University: SVV 260 549.

### Competing Interests

The authors declare that they have no competing interests.

### Author Contributions

- Jana Pourova conceived and designed the experiments, performed the experiments, analyzed the data, prepared figures and/or tables, and approved the final draft.
- Patricia Dias performed the experiments, analyzed the data, prepared figures and/or tables, and approved the final draft.
- Milan Pour analyzed the data, authored or reviewed drafts of the article, and approved the final draft.
- Silvia Bittner Fialová analyzed the data, authored or reviewed drafts of the article, and approved the final draft.
- Szilvia Czigle analyzed the data, authored or reviewed drafts of the article, and approved the final draft.
- Milan Nagy performed the experiments, analyzed the data, authored or reviewed drafts of the article, and approved the final draft.
- Jaroslav Tóth analyzed the data, authored or reviewed drafts of the article, and approved the final draft.

- Viktória Lilla Balázs analyzed the data, authored or reviewed drafts of the article, and approved the final draft.
- Adrienn Horváth analyzed the data, authored or reviewed drafts of the article, and approved the final draft.
- Eszter Csikós analyzed the data, authored or reviewed drafts of the article, and approved the final draft.
- Ágnes Farkas analyzed the data, authored or reviewed drafts of the article, and approved the final draft.
- Györgyi Horváth analyzed the data, authored or reviewed drafts of the article, and approved the final draft.
- Přemysl Mladěnka performed the experiments, analyzed the data, authored or reviewed drafts of the article, and approved the final draft.

## Data Availability

This is a review article.

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
