# Peer review of "Proposed mechanisms of action of herbal drugs and their biologically active constituents in the treatment of coughs: an overview"

_PeerJ, doi:10.7717/peerj.16096_

## Round 0.1 · original submission · Major Revisions

Dear authors, thank you for your submission.

At this moment, your manuscript requires extensive revision.

Please, refer to the reviewers' comments for further details.

Please, be careful with 'direct translations', misleading jargon and incorrect formulas and facts. Be sure to highlight in which way this review is important or valid and/or the motivation for this work.

Please, confirm that your resubmitted version present a well-developed and supported argument that is set out in the introduction and, unresolved questions or future enterprises are identified in the conclusion.

·

Basic reporting

There are small errors in chemical structures nº 6, 14, 18 (the absolute configuration of chiral center was not indicated) and 17 (there is a carbone with more than two groups at the plane).

I think that In the introduction, there should be a paragraph talking a little about the issue of the popular use of medicinal plants, the need to conduct phytochemical studies to support this use and enable the development of a rational system of safe and effective use of phytotherapy, based on scientifically proven data. For this is precisely the focus of this manuscript, so the introduction should draw the reader's attention to this issue.

in line 1094 there is this affirmation "Given the high saponin content, the saponins were assumed as the components most responsible 1095 for these effects [208]" - I think that is not possible to ensure that the major constituints are the ones that causes the pharmacological effect. I suggest not say it.

Experimental design

no comment

Validity of the findings

I think that in conclusion, there is a topic that wasn´t discussed in the text. I suggest explore this topic before use it in conclusion.

line 1230 "On the other hand, this complexity makes the identification of the main active substance difficult. Moreover, the contents of the herbal drugs can vary depending on a number of factors, which can complicate the analysis."

Additional comments

The article is excelent. I´d like congradulate the authors

Reviewer 2 ·

Basic reporting

1. I suggested reorganizing the information for easily understood by the reader. For example, the authors started to define cough (line 40) and after they explained the physiological mechanisms for cough (line 89) with this approach, the authors avoided any reading sequence.
2. In general, the review has an inadequate balance of information. For example, in the section on Anti-inflammatory Effects, the information is pronounced, meanwhile the Expectorant Effects section is more colloquial.
3. Some points are lacking in the text's structure: Discussion of expectorant effects and the Clinical evidence of the anti-inflammatory effects.
4. The authors described the clinical evidence of different products of medicinal plants on the expectorant effects, but in the end, they do not give any type of conclusion (line 164-267).
Line 164: “Expectorant effects in vivo were confirmed, for example, in some Glycyrrhiza spp. components, Hedera helix L. extract or an ointment containing menthol and camphene [6-8]”. In the same way, in line 211, they just describe the experiment, but they do not contribute anything.
5. The information in the Table 1 is not significant. The cough is a multifactorial response and physiological event and the authors do not display any information vital for the use of medicinal plants or
6. In figure 1, the authors mentioned the chemical structures of important plant constituents. However, some of them, are not mentioned in the text, and they are not related to the information exposed. The authors did not mention all the purposes of the chemical compounds in this review.
7. I suggested checking the main concepts in this review because the authors often used indistinctly to refer to herbal drugs or medicinal plants and phytotherapy or herbal therapy and it is not the same.
8. In lines 127-128, the authors explained “Thus, some plants and natural substances cannot be categorized in a simple way…” indicating the expectorant and antitussive effects. However, they decide to use this classification in this review. It is suggested to the authors to set the classification by structural or type chemical compounds (terpenes, diterpenes, kauranes, etc.,) or another and the potential effects on the cough. For this reason, they can propose chemical compounds and their possible effects.
9. There are some paragraphs that do not imply an academic study. It is so colloquial. It seems that focus on diffusion paper or superficial information:
a. Line 42: “In contrast to this useful physiological function, frequent or severe coughing can produce undesirable effects such as a distress of respiratory muscles, interrupted sleep, urinary incontinence, impaired healing of wounds after surgery, and constitutes a social barrier”.
b. Line 56: This review is intended for pharmacognosists, who are experts in botanical aspects and would like to extend their knowledge from the pharmacological point of view, as well as for physicians or pharmacists who are consulted by the general public, interested in herbal self-medication.
10. It is a scientific review; it is not convenient to use misunderstanding meaning as “the irritant receptors”. They do not exist. (Line 92).

Experimental design

1. In this review, I suggested to include in the Survey Methodology included:
a. The period of the survey in PubMed and Science Direct.
b. Criteria. Inclusion and exclusion of the terms or pharmacological studies used.

Validity of the findings

1. They do not propose significant conclusions.

Additional comments

Minor comments
1. Please check the grammar:
a. The former are unmyelinated and slower conducting.
b. In line 216, 20 minutes.
2. In the review, I suggested writing the Latin words in cursive. Regardless of the fact it is a scientific name or plant.

Annotated reviews are not available for download in order to protect the identity of reviewers who chose to remain anonymous.

---

## Round 0.2 · Minor Revisions

Congratulations! I am delighted to inform you that your manuscript is almost ready to be accepted for publication. Thank you for your submission and hard work. Please revise and proofread your manuscript, paying particular attention to Table 2 - Ensure accurate information and include appropriate references.

---

## Round 0.3 · Minor Revisions

Thank you for your revised version. Although your manuscript showed significant improvements, some details still need to be addressed. Please, refer to the reviewers report. Moreover, I please ask you to re-upload (besides the track changes manuscript MS Word doc) all the figures, tables etc, fully proofread and/or corrected.

·

Basic reporting

The authors made the corrections to the chemical structures that I had requested in the first evaluation. Such structures are correct in this version.

The remarks I had made about the introduction were taken care of by the authors.

The remarks I had made about the conclusion were taken care of by the authors.

Experimental design

The bibliographic basis raised by the authors for this writing this review are updated and very well substantiated.

Validity of the findings

The article is very well worded. The conclusions are supported by the arguments and data presented in the development of the theme.

Additional comments

All the flaws in the article that I had pointed out in the first review have been corrected by the authors.

Reviewer 2 ·

Basic reporting

This is an interesting review and the authors have improved the information and grammar.
1. The paper is generally well-written and structured. However, some paragraphs are verbiage. E.g. Lines 40-43, 64-68.
2. The authors do not reply to the specific comments from the reviewer.

Experimental design

1. Previously, it has requested all the criteria, inclusion, and exclusion terms for their review, because it is so important to describe with satisfactory and adequate information. The authors might examine the methodological information that sheds light on the issues addressed in this paper. To increase the impact and importance of the review, authors should ensure careful strategies.

Validity of the findings

In this review, the authors proposed mechanisms of action of medicinal plants and their biological components on coughs. They exposed the structures and pharmacological effects of expectorant, antitussive, and anti-inflammatory effects of plants or chemical compounds isolated for plants. However, in the clinical use paragraph, they showed that medicinal plant components are used in other biological systems, but this is not the objective of this paper.

Additional comments

1. Check in Line 29, Pharmacological principles of their action, however,
2. Check in line 304: Mentha × piperita

---

## Round 0.4 · Minor Revisions

Dear authors, please revise the reviewer comments and resubmit your manuscript. It is my believe that you should do a throughout proofreading before final approval.

Reviewer 2 ·

Basic reporting

This is the third review of the paper titled "Proposed Mechanisms of Action of herbal drugs and their
biologically active constituents in the treatment of coughs: overview".

The authors have improved the information to explain the possible mechanisms of action of medicinal plants or their constituents on cough.

I mentioned previously, that the authors explained quercetin's anti-inflammatory effects in other systems. The authors mentioned this effect on rheumatoid arthritis, polycystic ovary syndrome women, patients on hemodialysis, thalassemic patients, etc. In the same way, the authors described another compound trans-Anethole trithione (trisulfuric derivative of trans-anethole) for xerostomia treatment.
In other paragraphs, they wrote: "Based on Persian folk medicine, Foeniculum vulgare (together with Tribulus terrestris, Myrtus communis, and Tamarindus indica) were used to prepare vaginal suppositories for the therapy of bacterial vaginosis..." (line 698-700).
"Within 26 weeks, luteolin, quercetin, and rutin (10 mg, 7 mg, and 3 mg per 1 kg body weight, respectively) were administered per os to autistic children in which inflammation in the brain regions related to cognitive function was assumed"
But it is not the objective of this review. I suggested removing that information, it is not necessary.

Experimental design

The study design information has been improved. No comment.

Validity of the findings

This paper is really interesting because it provides recent information on the possible mechanisms of plant compounds and plants in the treatment of coughs.

Additional comments

In line 29, check:
"experience. Pharmacological principles of their action, however, are..."
After the word "action" place a semicolon, not a comma.
In line 728: change quercetine for quercetin. Luteoline for luteolin
In line 728: place a comma before de and "...quercetin, and..."
In line 728: place a comma before de and "...7 mg, and..."
In line 729: check per os

---

## Round 0.5 · accepted · Accept

Dear authors, your manuscript is now accepted for publication. The previous minor revisions highlighted the need for some proofreading before processing it any further. About the reviewers' comment, I must highlight that authors can always rebut accordingly, and if you think it adds value to your review, by all means you can keep it as it has links of connection to your work and is properly cited. Hoping you will consider PeerJ in further publications. Many congratulations and thank you for all your work.